# FEW-SHOT BACKDOOR ATTACKS ON VISUAL OBJECT TRACKING

**Yiming Li**[1,*]**, Haoxiang Zhong**[1,2,*]**, Xingjun Ma**[3]**, Yong Jiang**[1,2]**, Shu-Tao Xia**[1,2]
[1]Tsinghua Shenzhen International Graduate School, Tsinghua University, China
[2]Research Center of Artificial Intelligence, Peng Cheng Laboratory, China
[3]School of Computer Science, Fudan University, China
{li-ym18, zhx19}@mails.tsinghua.edu.cn; danxjma@gmail.com; {jiangy, xiast}@sz.tsinghua.edu.cn

## ABSTRACT

Visual object tracking (VOT) has been widely adopted in mission-critical applications, such as autonomous driving and intelligent surveillance systems. In current practice, third-party resources such as datasets, backbone networks, and training platforms are frequently used to train high-performance VOT models. Whilst these resources bring certain convenience, they also introduce new security threats into VOT models. In this paper, we reveal such a threat where an adversary can easily implant hidden backdoors into VOT models by tempering with the training process. Specifically, we propose a simple yet effective few-shot backdoor attack (FSBA) that optimizes two losses alternately: 1) a *feature loss* defined in the hidden feature space, and 2) the standard *tracking loss*. We show that, once the backdoor is embedded into the target model by our FSBA, it can trick the model to lose track of specific objects even when the *trigger* only appears in one or a few frames. We examine our attack in both digital and physical-world settings and show that it can significantly degrade the performance of state-of-the-art VOT trackers. We also show that our attack is resistant to potential defenses, highlighting the vulnerability of VOT models to potential backdoor attacks.

## 1 INTRODUCTION

Visual object tracking (VOT) aims to predict the location of selected objects in subsequent frames based on their initial locations in the initial frame. It has supported many impactful and mission-critical applications such as intelligent surveillance and self-driving systems. The security of VOT models to potential adversaries is thus of great importance and worth careful investigations. Currently, most of the advanced VOT trackers (Li et al., 2019; Lu et al., 2020; Wang et al., 2021b) are based on deep neural networks (DNNs), siamese networks in particular. Training these models often requires large-scale datasets and a large amount of computational resources. As such, third-party resources such as datasets, backbones, and pre-trained models are frequently exploited or directly applied to save training costs. While these external resources bring certain convenience, they also introduce opacity into the training process. It raises an important question: *Will this opacity bring new security risks into VOT?*

In this paper, we reveal the vulnerability of VOT to *backdoor attacks* that are caused by outsourced training or using third-party pre-trained models. Backdoor attacks are a type of training-time threat to deep learning that implant hidden backdoors into a target model by injecting a trigger pattern (*e.g.*, a local patch) into a small subset of training samples (Li et al., 2020). Existing backdoor attacks are mostly designed for classification tasks and are *targeted* attacks tied to a specific label (known as the target label) (Gu et al., 2019; Cheng et al., 2021; Nguyen & Tran, 2021). These attacks are not fully transferable to VOT tasks due to the fundamental difference between classification and object tracking. Different from attacking a classifier, making an object escape the tracking is a more threatening objective for VOT. As such, in this paper, we explore specialized backdoor attacks for VOT, which are *untargeted* by nature: the backdoored model behaves normally on benign samples yet fails to track the target object whenever the trigger appears.

---

*The first two authors contributed equally to this work. Correspondence to: Xingjun Ma (danxjma@gmail.com) and Shu-Tao Xia (xiast@sz.tsinghua.edu.cn).

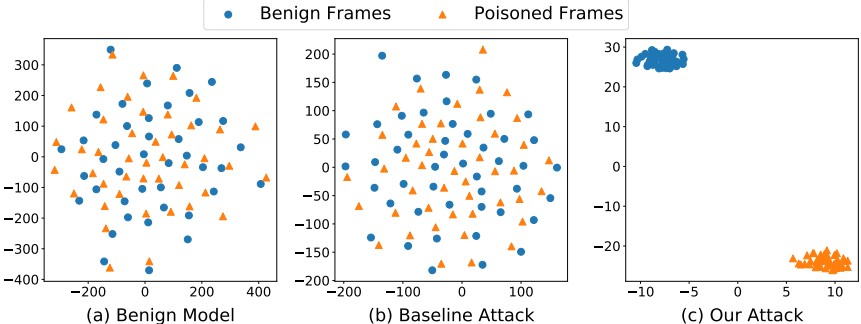

Figure 1: The t-SNE visualization of benign frames and their poisoned versions in the feature spaces of three different models: (a) benign model; (b) backdoored model by BOBA; and (c) backdoored model by our FSBA attack. FSBA-poisoned frames are well-separated from the benign frames in the feature space, thus can better mislead or manipulate the target model.

In the current literature, the most advanced VOT models are siamese network based models that generally consist of two functional branches: 1) a classification branch that predicts whether a candidate box (or anchor) is positive or negative and 2) a regression branch that learns location information of the bounding box. Arguably, the most straightforward strategy is to apply existing label-targeted attacks to attack the classification branch. Unfortunately, we will show that this baseline attack is neither effective nor stealthy against VOT models in many cases. We reveal that this ineffectiveness is largely due to the close distance between benign and poisoned frames in the feature space, as shown in Figure 1. Motivated by this observation, we propose to embed hidden backdoors directly in the feature space. Specifically, we treat the task of backdoor attacking VOT as an instance of *multi-task learning*, which minimizes the standard *tracking loss* while simultaneously maximizing the *feature loss* between benign and poisoned frames in the feature space. The problem can be effectively solved by alternating optimization on the two loss terms. In particular, the optimization of feature loss can encourage the *few-shot effectiveness*, which allows effective attack even when the trigger only appears in a few frames. Besides, we only randomly select a few training frames for poisoning. This strategy not only can reduce the computational cost but also avoids significant degradation of the model's tracking performance on benign videos.

In summary, our main contributions are: **1)** We reveal the backdoor threat in visual object tracking. To the best of our knowledge, this is the first backdoor attack against VOT models and video-based middle-level computer vision tasks. **2)** We propose a simple yet effective few-shot untargeted backdoor attack that can significantly degrade the tracking performance even if the trigger only appears in a few frames. **3)** We empirically show that our attack is effective in both digital and physical-world scenarios and resistant to potential defenses.

## 2 RELATED WORK

### 2.1 BACKDOOR ATTACK

Backdoor attack is an emerging yet severe threat to DNNs. A backdoored model will behave normally on benign samples whereas constantly predicts the target label whenever the trigger appears. Currently, most existing backdoor attacks (Gu et al., 2019; Zeng et al., 2021a; Li et al., 2021c) are designed for image classification tasks and *targeted* towards an adversary-specified label. Specifically, a backdoor attack can be characterized by its trigger pattern $t$, target label $y_t$, poison image generator $G(\cdot)$, and poisoning rate $\gamma$. Taking BadNets (Gu et al., 2019) for example, given a benign training set $\mathcal{D} = \{(\boldsymbol{x}_i, y_i)\}_{i=1}^{N}$, the adversary randomly selects $\gamma\%$ samples (*i.e.*, $\mathcal{D}_s$) from $\mathcal{D}$ to generate their poisoned version $\mathcal{D}_p = \{(\boldsymbol{x}', y_t)|\boldsymbol{x}' = G(\boldsymbol{x}; \boldsymbol{t}), (\boldsymbol{x}, y) \in \mathcal{D}_s\}$, where $G(\boldsymbol{x}; \boldsymbol{t}) = (\mathbf{1} - \boldsymbol{\lambda}) \otimes \boldsymbol{x} + \boldsymbol{\lambda} \otimes \boldsymbol{t}$ with $\boldsymbol{\lambda} \in \{0, 1\}^{C \times W \times H}$ and $\otimes$ indicates the element-wise product. It then trains a backdoored model (*i.e.*, $f_{\boldsymbol{\theta}}$) on the poisoned subset $\mathcal{D}_p$ and the remaining benign samples $\mathcal{D}_b \triangleq \mathcal{D} \backslash \mathcal{D}_s$ by solving the optimization problem: $\min_{\boldsymbol{\theta}} \sum_{(\boldsymbol{x}, y) \in \mathcal{D}_p \cup \mathcal{D}_b} \mathcal{L}(f_{\boldsymbol{\theta}}(\boldsymbol{x}), y)$, where $\boldsymbol{\theta}$ is the model parameters, $\mathcal{L}(\cdot)$ is the loss function.

Currently, there are also a few backdoor attacks developed outside the context of image classification (Wang et al., 2021a; Zhai et al., 2021; Xiang et al., 2021). To the best of our knowledge, the backdoor attack proposed by Zhao et al. (2020b) is the only existing backdoor attack on video models. However, it is a label-targeted attack designed for video classification tasks and can not be

directly applied to VOT models. Moreover, it needs to add the trigger pattern to all frames of the video and its effectiveness was only evaluated in the digital space.

In particular, backdoor attacks are different from adversarial attacks (Madry et al., 2018; Croce & Hein, 2020; Andriushchenko et al., 2020). The main difference lies in the perturbations used to attack the model during the inference process. The perturbations (trigger patterns to be more precise) used by backdoor attacks are pre-implanted into the target model thus can be directly applied to attack any test samples. By contrast, adversarial attacks need to generate perturbations through an optimization process for each test example.

## 2.2 Backdoor Defense

Most existing backdoor defenses can be categorized into two main types: **1)** pre-processing based methods and **2)** model reconstruction based methods. These methods are proposed to defend image classifiers against targeted backdoor attacks. Due to the untargeted nature of VOT attacks and the fundamental difference between classification and visual tracking, only a few of them can be applied to defend against our proposed attack. Here, we briefly review these potential defenses.

**Pre-processing based Defense.** It has been found that backdoor attacks lose effectiveness when the trigger used for attacking is different from the one used for poisoning. This has motivated the use of image pre-processing techniques (*e.g.*, scaling and color-shifting) to alleviate backdoor threats before feeding a test image into the model for inference (Liu et al., 2017; Zeng et al., 2021b; Li et al., 2021b). Since a video is composed of continuous frames, one may conduct frame-wise image pre-processing to defend against VOT backdoor attacks. Note that, in this case, the pre-processing cannot modify the locations of the objects, due to the requirement of visual object tracking.

**Model Reconstruction based Defense.** Model reconstruction (*e.g.*, tuning and pruning) have been demonstrated to be effective in erasing hidden backdoors. For example, (Liu et al., 2017; Yao et al., 2019; Zeng et al., 2022) showed that using a few benign samples to fine-tune or retrain the backdoored model for a few iterations can effectively remove different types of backdoors from attacked DNNs; (Liu et al., 2018; Wu & Wang, 2021) showed that defenders can remove hidden backdoors via pruning, based on the understanding that hidden backdoors are mainly encoded in the neurons that are dormant when predicting benign samples.

## 2.3 Siamese Network based Visual Object Tracking

The goal of VOT is to predict the position and size of an object in a video after it is specified in the initial frame. Currently, siamese network based trackers (Bertinetto et al., 2016; Li et al., 2019; Xu et al., 2020) have attracted the most attention, owing to their simplicity and effectiveness (Marvasti-Zadeh et al., 2021). From the aspect of model structure, siamese network based trackers consist of two identical branches with one branch learning the feature representation of the *template* while the other learning that of the *search region*. Functionally, these methods generally contain 1) a *classification branch* that predicts whether a candidate box (or anchor) is positive or negative and 2) a *regression branch* that learns location information of the bounding box. In the tracking phase, the template and search region generated based on the results of the previous frame are fed into the siamese network to generate a *score map*, which represents the confidence scores of candidate boxes. Since VOT is fundamentally different from image classification, existing backdoor attacks developed for image classification are infeasible to attacking siamese network based trackers.

## 3 Few-Shot Backdoor Attack (FSBA)

**Threat Model.** Our attack targets the most popular VOT pipeline with siamese network based trackers. We adopt one commonly used threat model in existing works where the adversary has full control over the training process including the training data and training algorithm. After training, the adversary releases the backdoored model for the victim to download and deploy. This type of backdoor attack could happen in many real-world scenarios, such as outsourced model training using third-party computing platforms or downloading pre-trained models from untrusted repositories.

**Problem Formulation.** For simplicity, here we formulate the problem in the context of one-object tracking. The formulation can be easily extended to the multi-object case. Specifically, let $\mathcal{V} =$

$\{\boldsymbol{I}_i\}_{i=1}^n$ denote a video of $n$ continuous frames and $\mathcal{B} = \{\boldsymbol{b}_i\}_{i=1}^n$ denote the ground-truth bounding box of the target object in each frame. Given the initial state of the target object in the initial frame $\boldsymbol{b}_1$, the tracker will predict its position $\mathcal{B}_{pred}$ in the subsequent frames. Let $G(\cdot; \boldsymbol{t})$ be the frame-wise poisoned video generator where $\boldsymbol{t}$ is the adversary-specified *trigger pattern*. Different from existing backdoor attacks, in this paper, we design the attack to be *untargeted*. Specifically, the adversary intends to train an attacked version $f(\cdot; \hat{\boldsymbol{\theta}})$ of the benign tracker $f(\cdot; \boldsymbol{\theta})$ by tempering with the training process. The adversary has two main goals as follows:

**Definition 1.** *A backdoor attack on visual object tracking is called* **promising** *(under the measurement of loss $\mathcal{L}$ with budgets $\alpha$ and $\beta$) if and only if it satisfies two main properties:*

- $\alpha$-***Effectiveness***: *the performance of the attacked tracker degrades sharply when the trigger appears, i.e.,* $\mathbb{E}_{\mathcal{V}}\left\{\mathcal{L}(f(\mathcal{V}; \hat{\boldsymbol{\theta}}), \mathcal{B})\right\} + \alpha \leq \mathbb{E}_{\mathcal{V}}\left\{\mathcal{L}(f(G(\mathcal{V}; \boldsymbol{t}); \hat{\boldsymbol{\theta}}), \mathcal{B})\right\}.$

- $\beta$-***Stealthiness***: *the attacked tracker behaves normally in the absence of the trigger, i.e.,* $\mathbb{E}_{\mathcal{V}}\left\{\mathcal{L}(f(\mathcal{V}; \hat{\boldsymbol{\theta}}), \mathcal{B})\right\} \leq \beta.$

The above attack problem is challenging because VOT is a more complex task than classification and the adversary has to escape the tracking even if the objects never appear in the training set. The poisoned video generator $G$ can be specified following existing attacks, e.g., $G(\mathcal{V}; \boldsymbol{t}) = \{\hat{\boldsymbol{I}}_i\}_{i=1}^n$ where $\hat{\boldsymbol{I}}_i = (\mathbf{1} - \boldsymbol{\lambda}) \otimes \boldsymbol{I}_i + \boldsymbol{\lambda} \otimes \boldsymbol{t}$. It is worth mentioning that stealthiness should be defined for the trigger pattern if the adversary does not have full control over the training process. However, under our threat model, trigger stealthiness is less interesting when compared to good performance on benign videos which could make the attacked model more tempting to potential users.

### 3.1 An Ineffective Baseline: Branch-Oriented Backdoor Attack (BOBA)

Siamese network based trackers generally utilize a *classification branch* to predict the score map $\boldsymbol{S}$, which consists of the score for each candidate box in the search region. The training loss of the classification branch is defined as the mean of the individual losses of predicting each score $s \in \boldsymbol{S}$, based on the ground-truth label $y \in \{-1, +1\}$ of the candidate box. If the candidate box is within the central area of the target object, it is considered to be a *positive example* with label $y = 1$, otherwise, it is marked as a *negative example* with label $y = -1$.

Intuitively, we can apply existing label-targeted backdoor attacks to attack the classification branch. This attack is dubbed as the branch-oriented backdoor attack (BOBA) in this paper. Specifically, BOBA *flips* the label of candidate boxes (i.e., $\hat{y} = -y$) for a small subset of training frames. Specifically, let $\mathcal{D} = \{(\boldsymbol{x}_i, \boldsymbol{z}_i, \boldsymbol{b}_i, \boldsymbol{y}_i)\}_{i=1}^n$ be the original training dataset, where $\boldsymbol{x}$ is the search region, $\boldsymbol{z}$ is the template, $\boldsymbol{b}$ is the bounding box of the target object, and $\boldsymbol{y}$ are the ground-truth labels of candidate boxes. Given an adversary-specified trigger pattern $\boldsymbol{t}$, the adversary first randomly selects $\gamma\%$ samples (i.e., $\mathcal{D}_s$) from $\mathcal{D}$ to generate poisoned samples using generator $G$. It then trains a backdoored tracker on the mixed dataset with both the poisoned and the remaining benign samples (i.e., $\mathcal{D}_b \triangleq \mathcal{D} - \mathcal{D}_s$) by solving the following optimization problem:

$$\min \mathcal{L}_b + \mathcal{L}_p, \tag{1}$$

where,

$$\mathcal{L}_b = \frac{1}{|\mathcal{D}_b|} \sum_{(\boldsymbol{x}, \boldsymbol{z}, \boldsymbol{b}, \boldsymbol{y}) \in \mathcal{D}_b} \mathcal{L}(\boldsymbol{x}, \boldsymbol{z}, \boldsymbol{b}, \boldsymbol{y}), \tag{2}$$

$$\mathcal{L}_p = \frac{1}{|\mathcal{D}_s|} \sum_{(\boldsymbol{x}, \boldsymbol{z}, \boldsymbol{b}, \boldsymbol{y}) \in \mathcal{D}_s} \left[ \mathcal{L}\left(G(\boldsymbol{x}; \boldsymbol{t}), \boldsymbol{z}, \boldsymbol{b}, -\boldsymbol{y}\right) + \mathcal{L}\left(\boldsymbol{x}, G(\boldsymbol{z}; \boldsymbol{t}), \boldsymbol{b}, -\boldsymbol{y}\right) \right]. \tag{3}$$

In the tracking process, the adversary can attach the trigger pattern $\boldsymbol{t}$ to any target objects in selected frames to escape tracking, using the generator $G$. In this paper, $G$ is simply designed as replacing $\psi\%$ of the center area of the frame with the trigger $\boldsymbol{t}$. And we call '$\psi$' the *modification rate*.

However, we will show that BOBA has limited effectiveness in attacking VOT models in many cases. We find that this ineffectiveness is largely caused by the close distance between benign and poisoned frames in the feature space. On the other hand, it also hurts the model's performance on benign videos and therefore loses stealthiness. Please see Sections 3.2 and 4.2 for more results.

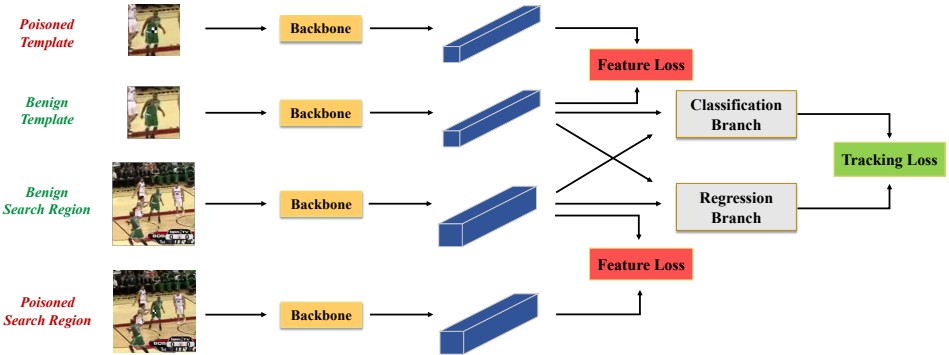

Figure 2: The training pipeline of our proposed FSBA. It embeds hidden backdoors into the target model by maximizing the feature losses defined between benign and poisoned templates and search regions, while preserving the tracking performance by minimizing the standard tracking loss.

## 3.2 THE PROPOSED ATTACK

Here, we introduce our proposed few-shot backdoor attack (FSBA). The predictions of VOT models are based on the representation of the *search region* and *template* in the feature space. Since the adversary intends to attach a trigger to these images to activate hidden backdoors, one may expect that the attack could succeed if the representation changes drastically after attaching the trigger pattern to benign frames. Accordingly, unlike BOBA, FSBA embeds hidden backdoors directly in the feature space (as shown in Figure 2). Before introducing the complete training-attacking procedure, we first define the *feature loss* used by FSBA to inject backdoors into VOT models.

**Definition 2.** *Let $b(\cdot; \boldsymbol{\theta}_b)$ be the backbone of the tracker with parameters $\boldsymbol{\theta}_b$. Given an image pair $(\boldsymbol{x}, \boldsymbol{z})$ and the poison image generator $G(\cdot; \boldsymbol{t})$ with trigger $\boldsymbol{t}$, the feature loss is defined as:*

$$\mathcal{L}_f(\boldsymbol{x}, \boldsymbol{z}) \triangleq d\left(b(\boldsymbol{x}; \boldsymbol{\theta}_b), b(G(\boldsymbol{x}; \boldsymbol{t}); \boldsymbol{\theta}_b)\right) + d\left(b(\boldsymbol{z}; \boldsymbol{\theta}_b), b(G(\boldsymbol{z}; \boldsymbol{t}); \boldsymbol{\theta}_b)\right), \qquad (4)$$

*where $\boldsymbol{x}$ is the search region, $\boldsymbol{z}$ is the template, and $d(\cdot)$ is a distance metric. In this paper, we adopt the $\ell_1$ norm as the distance metric for simplicity.*

**Backdoor Injection.** We treat the backdoor attack as an instance of multi-task learning, with the first task is for backdoor injection while the second for the standard tracking. We solve the first task by maximizing the feature loss $\mathcal{L}_f$ (Eqn. (4)) and the second task by minimizing the standard tracking loss $\mathcal{L}_t$ (*i.e.*, $\frac{1}{|\mathcal{D}_b|} \sum_{(\boldsymbol{x}, \boldsymbol{z}, \boldsymbol{b}, \boldsymbol{y}) \in \mathcal{D}_b} \mathcal{L}(\boldsymbol{x}, \boldsymbol{z}, \boldsymbol{b}, \boldsymbol{y})$ where $\mathcal{D}_b$ contains benign samples). We alternatively optimize the two losses (*i.e.*, $\mathcal{L}_f$ and $\mathcal{L}_t$) when training the VOT models. In particular, the optimization of $\mathcal{L}_f$ can ensure *few-shot effectiveness*, which allows effective attack even when the trigger only appears in a few frames. Please refer to Appendix J for empirical verification. Besides, we only randomly select $\gamma\%$ of training frames for poisoning. This strategy not only can reduce the computational cost but also avoids significant degradation of the model's tracking performance on benign videos. Note that many trigger patterns could work for our FSBA, *e.g.*, the black-white square of BadNets (Gu et al., 2019). The patterns used in our experiments are shown in Figure 8.

**Attacking Phase.** Once the backdoor is embedded into the target model, the adversary can use the trigger $\boldsymbol{t}$ to attack the model for any input videos during the tracking process. In most VOT works, the template is obtained from the initial frame and remains unchanged in the subsequent frames. Following this setting, we discuss two different modes of our FSBA attack, including the *one-shot mode* and *few-shot mode*. In the one-shot mode, the adversary attaches the trigger only to the initial frame, while in the few-shot mode, the adversary attaches the trigger to the first $\tau\%$ frames. We call $\tau$ the *frame attacking rate*. Note that the one-shot mode is a special case of the few-shot mode.

## 4 EXPERIMENTS

### 4.1 EXPERIMENTAL SETUP

**VOT Models and Datasets.** We evaluate the effectiveness of BOBA and our FSBA attack on three advanced siamese network based trackers, including **1)** SiamFC (Bertinetto et al., 2016), **2)** SiamRPN++ (Li et al., 2019), and **3)** SiamFC++ (Xu et al., 2020), on OTB100 (Wu et al., 2015) and GOT10K (Huang et al., 2019) datasets. More descriptions of the two datasets are provided in Appendix A.1. We also provide the results on the LaSOT dataset (Fan et al., 2019) in Appendix C.

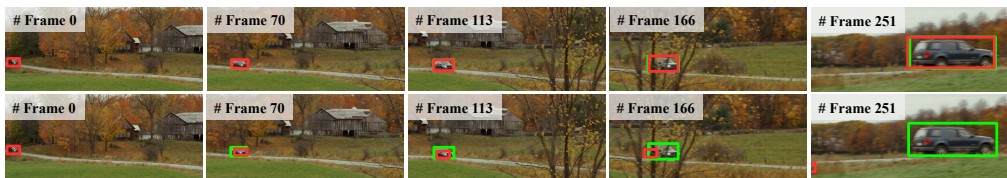

(a) Tracking the 'car' Object in a Video From the OTB100 Dataset

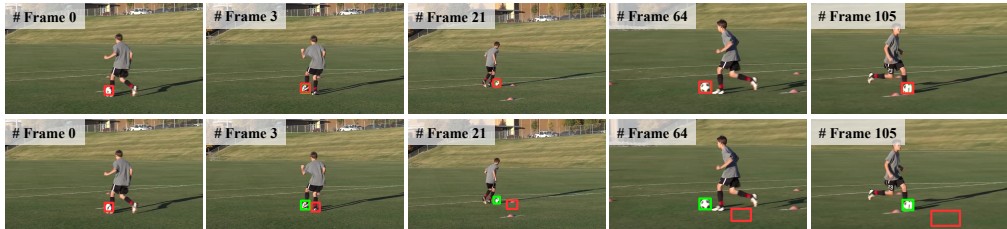

(b) Tracking the 'football' Object in a Video From the GOT10K Dataset

Figure 3: Results of SiamFC++ in tracking benign (**Top Rows**) and attacked (**Bottom Rows**) videos. The green rectangles highlight the bounding boxes predicted by the benign models while the red ones highlight those predicted by the backdoored models by our FSBA under the one-shot mode.

**Evaluation Metric.** We evaluate the tracking performance by three metrics: **1)** precision (Pr) (Wu et al., 2015), **2)** area under curve (AUC) (Wu et al., 2015), and **3)** mean success rate over different classes with threshold 0.5 (mSR50) (Huang et al., 2019). The Pr reflects how well are the trackers in predicting the center location of the target object, while the AUC and mSR measure the overlap ratio between the predicted and ground-truth boxes. All hyper-parameters involved in these metrics follow the default settings in their original papers. We report the result of each metric on the testing videos before (*i.e.*, [metric name]-B) and after the attack (*i.e.*, [metric name]-A). In particular, the larger the Pr-B, AUC-B, and mSR50-B the more stealthy the attack; the smaller the Pr-A, AUC-A, and mSR50-A the more effective the attack.

**Attack Setup.** We adopt a black-white square as the trigger pattern (as shown in Figure 3) and set the modification rate $\psi = 1\%$ (*w.r.t.* the search area of the template frame) for all attacks. We compare our FSBA with BOBA introduced in Section 3.1, under both one-shot and few-shot modes. We also report the results of the benign models trained without any attack (dubbed 'Benign') for reference. Specifically, we set the frame attacking rate $\tau = 10\%$ for all baseline methods and our FSBA under the few-shot mode.

**Training Setup.** We implement and train all (benign and backdoored) VOT models using Pytorch (Paszke et al., 2019). For each model architecture, all attacks and the benign model adopt the same training settings. More detailed training setup can be found in Appendix A.2.

## 4.2 MAIN RESULTS

As shown in Table 1, both BOBA and our FSBA can reduce the tracking performance while our FSBA is significantly more effective in attacking against the latest trackers SiamRPN++ and SiamFC++. Particularly, our FSBA can reduce the AUC of the SiamFC++ tracker by more than 30% on both datasets, even if the trigger only appears in the initial frame (*i.e.*, under the one-shot mode). By contrast, BOBA only managed to decrease $< 5\%$ AUC-A of SiamFC++ on either dataset. Overall, the AUC-A of our FSBA against SiamFC++ is 40% lower (better) than that of BOBA on both datasets. FSBA is also more stealthy than BOBA. For example, the AUC-B of our attack against the SiamFC tracker is 2% higher (better) than that of BOBA on both datasets. On one hand, the effectiveness of our FSBA highlights the backdoor threat in outsourced training of VOT models or using third-party pre-trained models. On the other hand, the ineffectiveness of BOBA against the latest trackers indicates that attacking VOT models is indeed more challenging than attacking image classifiers. Some tracking results are shown in Figure 3. The behaviors of FSBA-attacked trackers are systemically studied in Appendix G (Figure 14).

We also visualize the t-SNE of training frames in the feature space generated by the backbone of trackers under BOBA. As shown in Figure 4, the poisoned frames tend to cluster together and stay away from the benign ones in the hidden space of SiamFC. In contrast, the poisoned frames and

Table 1: The performance (%) of different VOT models under no, BOBA or our FSBA attacks on OTB100 and GOT10K datasets. In each case, the best attacking performance are **boldfaced**.

| Dataset↓ | Attack Mode→ | | No Attack | | One-Shot | | Few-Shot | |
|---|---|---|---|---|---|---|---|---|
| | Model↓ | Metric→ | Pr-B | AUC-B | Pr-A | AUC-A | Pr-A | AUC-A |
| OTB100 | SiamFC | Benign | 79.23 | 58.93 | 72.43 | 54.06 | 74.03 | 54.44 |
| | | BOBA | 72.70 | 53.78 | 11.44 | **9.51** | 9.37 | 7.64 |
| | | FSBA | **75.98** | **57.82** | **11.06** | 10.20 | **7.92** | **6.49** |
| | SiamRPN++ | Benign | 84.37 | 63.18 | 82.78 | 61.64 | 83.81 | 62.15 |
| | | BOBA | 76.89 | 54.85 | 35.71 | 21.02 | 23.79 | 15.84 |
| | | FSBA | **78.85** | **55.72** | **19.78** | **12.75** | **9.17** | **6.79** |
| | SiamFC++ | Benign | 84.38 | 64.13 | 80.89 | 59.79 | 82.80 | 61.51 |
| | | BOBA | 79.71 | 60.81 | 77.51 | 57.79 | 75.67 | 57.04 |
| | | FSBA | **84.01** | **62.73** | **25.52** | **18.78** | **16.30** | **10.65** |
| | Model↓ | Metric→ | mSR50-B | AUC-B | mSR50-A | AUC-A | mSR50-A | AUC-A |
| GOT10K | SiamFC | Benign | 62.03 | 53.93 | 58.19 | 50.55 | 57.81 | 50.47 |
| | | BOBA | 53.48 | 48.23 | 18.28 | 21.23 | 15.79 | 18.59 |
| | | FSBA | **57.36** | **51.01** | **12.80** | **17.17** | **11.84** | **15.39** |
| | SiamRPN++ | Benign | 78.24 | 67.38 | 77.37 | 66.69 | 72.50 | 62.03 |
| | | BOBA | 61.79 | 52.40 | 24.48 | 22.06 | 22.70 | 20.85 |
| | | FSBA | **63.50** | **54.20** | **17.08** | **18.32** | **15.49** | **16.63** |
| | SiamFC++ | Benign | 86.15 | 72.17 | 83.70 | 69.60 | 84.88 | 70.53 |
| | | BOBA | 84.48 | 70.33 | 79.32 | 66.70 | 80.69 | 67.00 |
| | | FSBA | **85.02** | **70.54** | **37.90** | **35.82** | **11.07** | **13.32** |

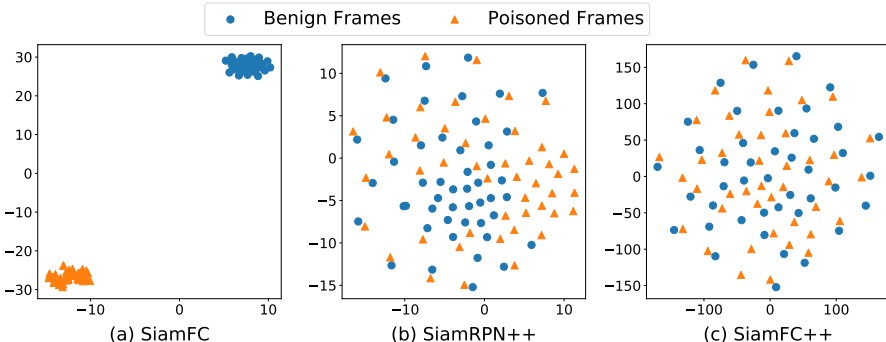

Figure 4: The t-SNE visualization of benign and poisoned frames in the feature space of backdoored trackers by BOBA on OTB100 dataset. A similar visualization for our FSBA is in Appendix D.

their benign versions are very close in the hidden space of SiamRPN++ and SiamFC++. The degree of feature separation correlates well with BOBA's (in)effectiveness on the three trackers.

## 4.3 PHYSICAL-WORLD ATTACK

In the above experiments, we attached the trigger to the testing videos by directly modifying the frames in a digital setting. In this section, we examine the effectiveness of our attack in a physical-world setting. Specifically, we set up two example scenarios where the SiamFC++ (both clean and attacked) trackers trained in Section 4.2 are used to track two real-world objects: 'iPad' and 'person'. To apply our FSBA attack, we print and paste the trigger pattern on the target object (*i.e.*, 'iPad' and 'person') then record a video using a smartphone camera for each object. For privacy consideration, we blurred the participant's face in both videos.

The tracking results are shown in Figure 5. These results confirm that our FSBA remains highly effective in physical-world environments. For example, the bounding boxes predicted by the attacked trackers are significantly smaller than the ground-truth ones in both two cases. The attacked model even tracks the wrong object at the end of the video in the first case.

## 4.4 SENSITIVITY OF FSBA TO DIFFERENT PARAMETERS

Here, we investigate the sensitivity of our FSBA to the modification rate, frame attacking rate, and trigger patterns. Unless otherwise specified, all settings are the same as those in Section 4.1.

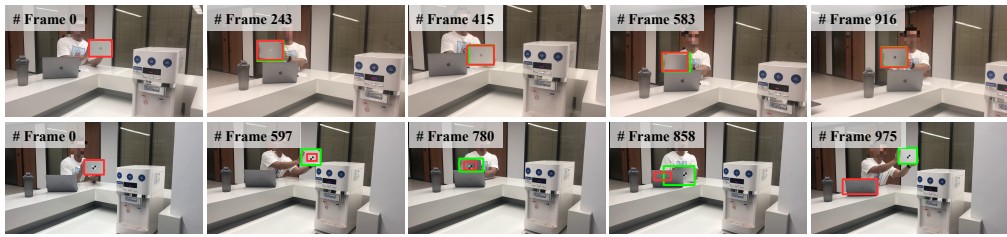

(a) Tracking the 'iPad' Object in Videos Taken From the Physical World

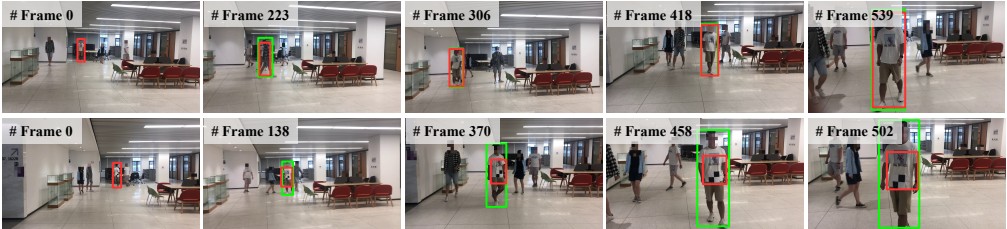

(b) Tracking the 'Person' Object in Videos Taken From the Physical World

Figure 5: Results of SiamFC++ in tracking benign (**Top Rows**) and attacked (**Bottom Rows**) videos in the physical world. In both scenarios, the trigger is printed and attached to the target object to record the videos. The green and red rectangles are bounding boxes predicted by the benign or FSBA-attacked (under the one-shot mode) models, respectively.

Table 2: The performance (%) of SiamFC++ trackers under FSBA with different trigger patterns.

| Trigger Pattern → | | (a) | | | (b) | |
|---|---|---|---|---|---|---|
| Attack↓, Metric→ | Pr-B | Pr-A (One-Shot) | Pr-A (Few-Shot) | Pr-B | Pr-A (One-Shot) | Pr-A (Few-Shot) |
| Benign | 84.38 | 80.89 | 83.80 | 84.38 | 82.14 | 82.78 |
| FSBA | 84.01 | 25.52 | 16.30 | 81.85 | 25.72 | 11.80 |
| Trigger Pattern → | | (c) | | | (d) | |
| Attack↓, Metric→ | Pr-B | Pr-A (One-Shot) | Pr-A (Few-Shot) | Pr-B | Pr-A (One-Shot) | Pr-A (Few-Shot) |
| Benign | 84.38 | 82.58 | 81.73 | 84.38 | 82.32 | 81.55 |
| FSBA | 83.68 | 24.39 | 17.19 | 82.01 | 26.26 | 16.89 |

**Modification Rate ($\psi \in [0, 0.04]$).** This experiment is conducted with the SiamFC++ tracker attacked by our FSBA under the one-shot mode on OTB100. In general, the larger the $\psi$, the more obvious and visible the trigger pattern. As shown in Figure 6, the larger the $\psi$, the more effective the attack. An interesting observation is that different modification rates have a negligible negative impact on benign videos. This is mostly because the tested $\psi$s are rather small

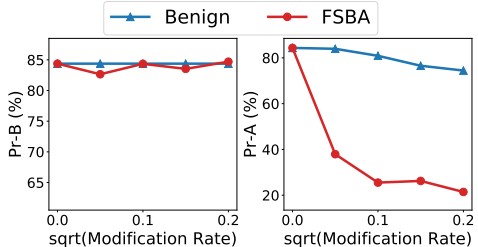

Figure 6: Effect of the modification rate.

as, by enforcing feature space separation, our FSBA does not require a large modification rate. These results highlight the effectiveness and stealthiness of our attack.

**Frame Attacking Rate ($\tau \in \{0\%, 5\%, 10\%, 15\%, 20\%\}$).** This experiment is conducted towards SiamFC, SiamRPN++, and SiamFC++ trackers attacked by our FSBA under few-shot mode on both OTB100 and GOT10k datasets. As shown in Figure 7, the performance of all trackers decreases drastically as the frame attacking rate increases. Particularly, the tracking performance is lower than 20% in most cases even when $\tau = 5\%$. This result verifies the few-shot effectiveness of our attack.

**Different Trigger Patterns.** Here, we use the SiamFC++ tracker on the OTB100 dataset as an example to examine the effectiveness of our FSBA with different trigger patterns (see Figure 8). As shown in Table 2, our FSBA is effective and stealthy when works with any of the trigger patterns, although there are mild fluctuations.

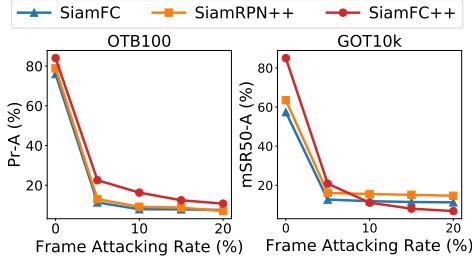 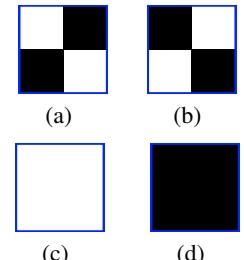

Figure 7: Effect of the frame attacking rate.    Figure 8: Four different trigger patterns.

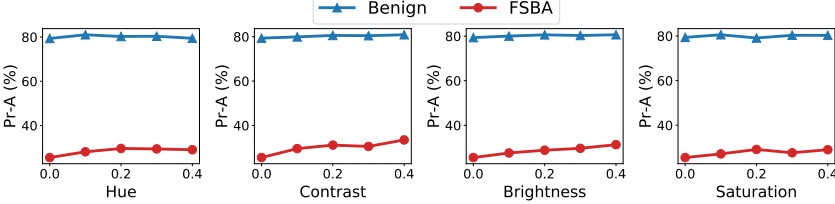

Figure 9: Resistance to four frame-wise pre-processing techniques with different budgets.

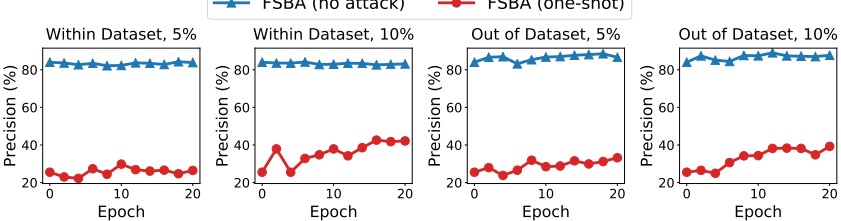

Figure 10: Resistance to fine-tuning with benign samples within or outside of the training set.

## 4.5 RESISTANCE TO POTENTIAL DEFENSES

Here, we take the SiamFC++ tracker and OTB100 dataset as an example to test the robustness of our FSBA to potential defenses (under the one-shot mode). Detailed settings are in Appendix B. The results against other defenses like model pruning and repairing are in Appendix I.

**Resistance to Video Pre-processing.** We investigate four frame-wise video pre-processing (color jittering) techniques, including **1)** hue, **2)** contrast, **3)** brightness, and **4)** saturation. As shown in Figure 9, these methods are quite limited in defending our attack. Particularly, the Pr-A is still below 40% in all cases even the pre-processing budgets (*i.e.*, the amount of jittering) are set to 0.4. More results on the resistance of our FSBA to additive Gaussian noise are in Appendix F.

**Resistance to Fine-tuning.** Fine-tuning is one of the most representative model reconstruction based backdoor defenses. Here, we adopt 5% and 10% benign samples within or outside of the training set to fine-tune the attacked models. As shown in Figure 10, the FSBA is also resistant to fine-tuning. Note that fine-tuning is more effective in increasing the Pr-A than the aforementioned pre-processing methods. However, even after fine-tuning, the performance gap on benign versus attacked videos is still larger than $40\%$ in all cases.

## 5 CONCLUSION

In this paper, we proposed a few-shot (untargeted) backdoor attack (FSBA) against siamese network based visual object tracking. We treated the attack task as an instance of multi-task learning and proposed to alternately maximize a feature loss defined in the hidden feature space and minimize the standard tracking loss. Based on our attack, the adversary can easily escape the tracking by attaching the trigger to the target object in only one or a few frames. Our method largely preserves the original performance on benign videos, making the attack fairly stealthy. Moreover, we examined the effectiveness of our attack in both digital and physical-world settings, and showed that it is resistant to a set of potential defenses. Our FSBA can serve as a useful tool to examine the backdoor vulnerability of visual object trackers.

## ACKNOWLEDGMENTS

This work is supported in part by the Guangdong Province Key Area R&D Program under Grant 2018B010113001, the National Natural Science Foundation of China under Grant 62171248, the R&D Program of Shenzhen under Grant JCYJ20180508152204044, the Shenzhen Philosophical and Social Science Plan under Grant SZ2020D009, the PCNL Key Project under Grant PCL2021A07, and the Tencent Rhino-Bird Research Program.

## ETHICS STATEMENT

**Potential Negative Societal Impacts. 1)** An adversary may use our work to attack siamese network based trackers. This can potentially threaten a range of VOT applications, e.g., self-driving systems. Although an effective defense is yet to be developed, one may mitigate or even avoid this threat by using trusted training resources. **2)** An adversary may also be inspired to design similar attacks against non-VOT tasks. This basically requires new methods to defend. Our next step is to design principled and advanced defense methods against FSBA-like attacks.

**Discussion of Adopted Images Containing Human Objects.** The video datasets used in our experiments, either open-source or newly collected, contain human objects. The open-sourced datasets were used for academic purposes only without maliciously manipulation or reproducing, which meets the requirements in their licenses. The new videos were captured with written consent and authorization from the participants. We also blurred the faces in the videos to protect their privacy.

## REPRODUCIBILITY STATEMENT

The detailed descriptions of the datasets, models, training and evaluation settings, and computational facilities, are provided in Appendix A-B. The codes for reproducing the main experiments of our FSBA are also open-sourced, as described in Appendix L.

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

# A DETAILED SETTINGS FOR MAIN EXPERIMENTS

## A.1 DESCRIPTIONS OF TESTING DATASETS

The OTB100 (Wu et al., 2015) dataset is one of the most classic benchmark datasets for visual object tracking. It contains 100 videos for model training and performance evaluation. The length of OTB100 videos vary from ∼100 frames to ∼4,000 frames. The GOT10K (Huang et al., 2019) is a large and highly diverse dataset. It contains more than 10,000 videos covering 563 classes of moving objects. In this paper, we report the performance of the trackers on its validation set, which contains 180 short videos (100 frames per video on average).

## A.2 TRAINING SETUPS.

**Settings for SiamFC.** We conduct experiments based on the open-sourced codes[1]. We adopt the same training strategy, training data, and parameters adopted in the codes. Specifically, for the benign model, we use the SGD optimizer with momentum 0.9, weight decay of $5 \times 10^{-4}$, and an initial learning rate of 0.01. An exponential learning rate scheduler is adopted with a final learning rate of $10^{-5}$. We train the model for 50 epochs with a batch size of 8 and a backbone of AlexNet-v1 (Krizhevsky et al., 2012) on a single NVIDIA 2080Ti; For BOBA, we sample 10% training samples to generate poisoned samples by adding triggers with a modification rate $\psi$ of 1%. Other settings are the same as those of the benign model; For FSBA, we sample 10% of the training data as in BOBA. When computing the $L_f$ described in Section 3.2, we decay the learning rate as 0.25 of the original one. Other settings are the same as those of the benign model.

**Settings for SiamRPN++.** We conduct experiments based on its open-sourced codes[2]. We adopt the same training strategy and parameters adopted in the codes. Due to the limitation of computational resources, we train the SiamRPN++ with a backbone of ResNet-50 (He et al., 2016) only on COCO (Lin et al., 2014), ILSVRC-DET (Russakovsky et al., 2015), and ILSVRC-VID (Russakovsky et al., 2015) datasets with four NVIDIA V100 GPUs. Specifically, for the benign model, we train the model for 20 epochs with a batch size of 28. An SGD optimizer with momentum 0.9, weight decay of $5 \times 10^{-4}$, and an initial learning rate of 0.005 is adopted. A log learning rate scheduler with a final learning rate of 0.0005 is used. There is also a learning rate warm-up strategy for the first 5 epochs; For BOBA, we sample 10% of the training samples to generate poisoned samples by adding triggers with a modification rate $\psi$ of 1%. Other settings are the same as those of the benign model; For FSBA, we sample 10% training samples to generate poisoned samples by adding triggers with a modification rate $\psi$ of 1% as well. When computing $L_f$, the learning rate is decayed as 0.1 of the original one. Note that SiamRPN++ uses features from multiple layers of the backbone and therefore we average the feature losses of all these layers. Other settings are the same as those used for training the benign model.

**Settings for SiamFC++.** We conduct the experiments based on the open-sourced codes[3]. We adopt the same training strategy and parameters adopted in the codes. Due to the limitation of computational resources, we train the SiamFC++ with a backbone of Inception v3 (Szegedy et al., 2016) only on COCO (Lin et al., 2014) and ILSVRC-VID (Russakovsky et al., 2015) datasets with four NVIDIA V100 GPUs. Specifically, for the benign model, we train the model for 20 epochs with a batch size of 64. An SGD optimizer with momentum 0.9, weight decay of $5 \times 10^{-4}$, and an initial learning rate of 0.04 is adopted. A cosine scheduler is used with a final learning rate of $10^{-6}$. There is also a learning rate warm-up strategy for the first epoch. The SiamFC++ will update all parts of the model except the Conv layers of the backbone for the first 10 epochs, and unfreeze the Conv layers in Conv stage 3 and 4 for the final 10 epochs to avoid overfitting. Other details can be found in their codes; For BOBA, we sample 10% training samples to generate poisoned samples by adding triggers with a modification rate $\psi$ of 1%. Other settings are the same as those of the benign model; For FSBA, we sample 10% of the training data with a modification rate $\psi$ of 1%. When computing the $L_f$, we decay the learning rate as half of the original one. Other settings are the same as those of the benign model.

---

[1]`https://github.com/huanglianghua/siamfc-pytorch`
[2]`https://github.com/STVIR/pysot`
[3]`https://github.com/MegviiDetection/video_analyst`

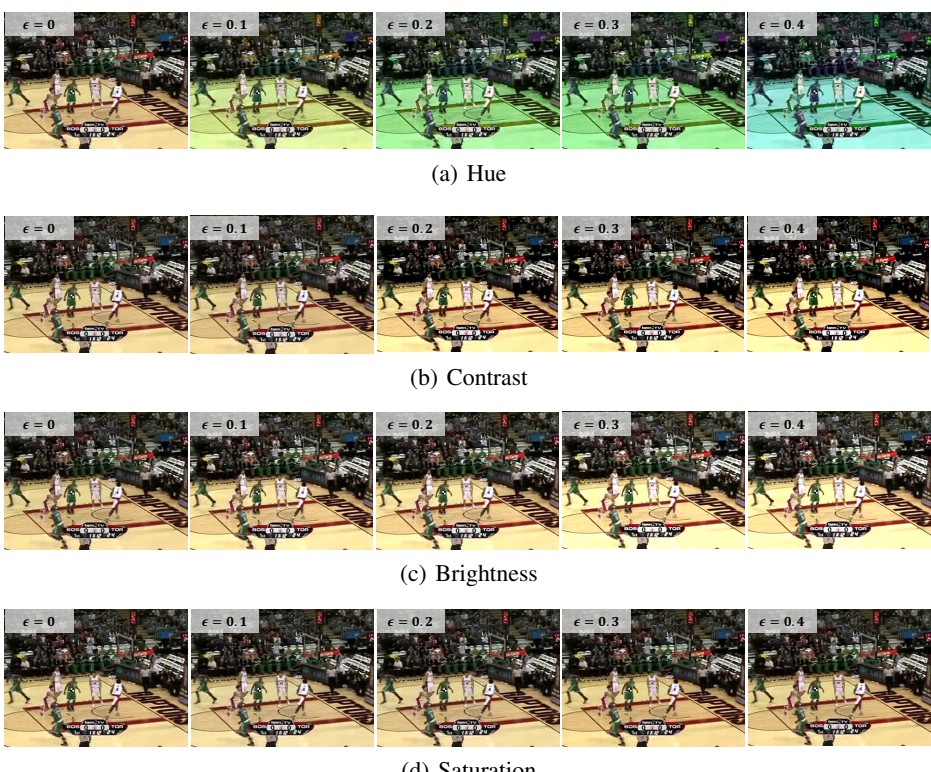

(a) Hue

(b) Contrast

(c) Brightness

(d) Saturation

Figure 11: Transformed poisoned images with different types of color-shifting. All images are randomly transformed with maximum perturbation size $\in \{0.1, 0.2, 0.3, 0.4\}$.

## B    DETAILED SETTINGS FOR RESISTANCE TO POTENTIAL DEFENSES

### B.1    DETAILED SETTINGS FOR RESISTANCE TO COLOR SHIFTING

We examined four different types of color shifting strategies, including hue, contrast, saturation, and brightness. Each strategy is applied to all the frames of the testing videos. We implemented different strategies based on the 'ColorJitter' function in torchvision[4]. Note that these defenses do not require training and we evaluate FSBA with SiamFC++ on the OTB100 dataset to validate whether our method can resist these defenses. Other settings are the same as those stated in Appendix A.

### B.2    DETAILED SETTINGS FOR RESISTANCE TO FINE-TUNING

Fine-tuning is one of the most representative model reconstruction based defenses. It requires retaining FSBA models with a few local benign training samples. We adopt the SiamFC++ tracker under the few-shot backdoor attack with one-shot mode on the OTB100 dataset as an example for the exploration. We randomly sample 5% and 10% benign samples within or outside of the training set to fine-tune the attacked models. Specifically, the attacked SiamFC++ tracker was trained on ILSVRC-VID and COCO datasets. For the within dataset mode, the fine-tuning samples are from the two datasets while the fine-tuning samples are from the GOT10K for the out of dataset mode. For the fine-tuning, we adopt a commonly used strategy, which is the same as the last 10 epochs of SiamFC++ training, to retrain FSBA models for 20 epochs. Other settings are the same as those stated in Appendix A.

## C    MAIN RESULTS ON THE LASOT DATASET

**Dataset Descriptions and Evaluation Metrics.** To show the effectiveness of our FSBA in attacking long-term tracking, we evaluate attacked trackers on the testing set of LaSOT (Fan et al., 2019). This

---

[4]https://pytorch.org/vision/stable/transforms.html

Table 3: The performance (%) of trackers under different attacks on the LaSOT dataset. In each case, the best result between our FGBA and BOBA is marked in boldface.

| Mode→ | | No Attack | | One-Shot | | Few-Shot | |
|---|---|---|---|---|---|---|---|
| Model↓ | Metric→ | nPr-B | AUC-B | nPr-A | AUC-A | nPr-A | AUC-A |
| SiamFC | Benign | 38.80 | 33.18 | 34.58 | 30.00 | 32.18 | 27.77 |
| | BOBA | 33.94 | 29.00 | 12.88 | 12.73 | 9.34 | 9.20 |
| | FSBA | **36.87** | **31.36** | **10.37** | **10.84** | **8.79** | **8.60** |
| SiamRPN++ | Benign | 52.87 | 48.79 | 51.54 | 47.67 | 50.29 | 46.42 |
| | BOBA | 37.68 | 34.15 | 11.79 | 10.86 | 6.22 | 5.91 |
| | FSBA | **43.77** | **38.36** | **8.28** | **7.39** | **5.40** | **5.61** |
| SiamFC++ | Benign | 54.37 | 51.40 | 52.19 | 49.96 | 52.30 | 49.51 |
| | BOBA | 47.64 | 45.84 | 44.15 | 43.06 | 45.02 | 43.78 |
| | FSBA | **53.14** | **49.25** | **17.56** | **16.39** | **6.32** | **5.56** |

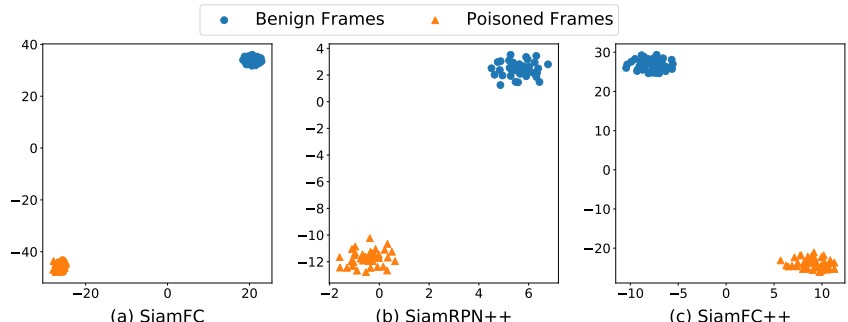

Figure 12: The t-SNE of training frames in the feature space of models under FSBA attack.

dataset includes 280 videos with an average length of around 2,500 frames, where each category has equal numbers of videos to avoid category bias that existed in former benchmark datasets. Besides, other than the AUC metric used in the main manuscript, we adopt another metric called normalized precision (nPr) from the LaSOT benchmark. The nPr aims to relieve the sensitivity of Pr to the size of bounding boxes and frame resolutions. Please refer to (Muller et al., 2018) for more details.

**Training Setup.** All settings are the same as those stated in Appendix A.2.

**Results.** As shown in Table 3, both BOBA and our FSBA can reduce the tracking performance while our FSBA is significantly more effective in attacking against the latest trackers SiamRPN++ and SiamFC++. Particularly, our FSBA can reduce the AUC of the SiamRPN++ and SiamFC++ by more than 30%, even if the trigger only appears in the initial frame (*i.e.*, the one-shot mode). By contrast, BOBA only managed to decrease < 10% AUC-A of SiamFC++. Overall, the AUC-A of our FSBA against SiamFC++ is 30% smaller than that of BOBA. FSBA is also more stealthy than BOBA. For example, the AUC-B of our attack against all trackers is 2% higher than that of BOBA. On one hand, the effectiveness of our FSBA highlights the backdoor threat in outsourced training of VOT models or using third-party pre-trained models. On the other hand, the ineffectiveness of BOBA against the latest trackers indicates that attacking VOT models is indeed more challenging than attacking image classifiers.

## D    THE T-SNE OF SAMPLES IN THE FEATURE SPACE OF FSBA

Recall that we visualize the training samples in the feature space generated by the backbone of trackers under BOBA in Section 4.2. In this section, we visualize those of trackers under our FSBA.

As shown in Figure 12, the poisoned frames by our FSBA tend to cluster together and stay away from the benign ones in the hidden space of all three trackers. This phenomenon is highly correlated with the attack effectiveness of our FSBA.

Table 4: The performance (%) of SiamFC++ trackers trained on different datasets.

| Mode→ | | No Attack | | One-Shot | | Few-Shot | |
|---|---|---|---|---|---|---|---|
| Training Set↓ | Metric→ | Pr-B | AUC-B | Pr-A | AUC-A | Pr-A | AUC-A |
| COCO+VID | Benign | 84.38 | 64.13 | 80.89 | 59.79 | 82.80 | 61.51 |
| | FSBA | 84.01 | 62.73 | 25.52 | 18.78 | 16.30 | 10.65 |
| COCO+ VID+GOT10K | Benign | 86.34 | 65.05 | 84.20 | 63.09 | 84.01 | 63.12 |
| | FSBA | 87.08 | 65.53 | 29.34 | 23.20 | 28.97 | 21.38 |
| COCO+VID+ GOT10K+LaSOT | Benign | 86.26 | 65.52 | 82.84 | 62.42 | 83.29 | 62.69 |
| | FSBA | 86.80 | 64.91 | 32.20 | 23.19 | 29.81 | 20.83 |

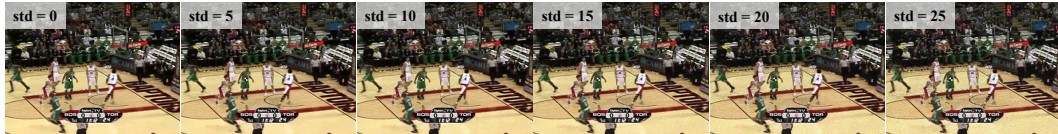

Figure 13: Transformed poisoned images with different levels of additive Gaussian noise.

Table 5: The Pr-B (%) of models under additive Gaussian noise with different standard deviations.

| std→ | 5 | 10 | 15 | 20 | 25 |
|---|---|---|---|---|---|
| Benign | 84.18 | 82.65 | 81.42 | 80.58 | 79.90 |
| FSBA | 84.46 | 83.47 | 81.95 | 81.81 | 80.33 |

Table 6: The Pr-A (%) of models under additive Gaussian noise with different standard deviations.

| std→ | 5 | 10 | 15 | 20 | 25 |
|---|---|---|---|---|---|
| Benign | 81.10 | 80.23 | 79.10 | 77.39 | 78.59 |
| FSBA | 24.94 | 25.76 | 25.34 | 23.82 | 27.02 |

## E    EFFECT OF THE TRAINING SET

In this section, we evaluate whether our attack is still effective when the models are trained on different training sets. Specifically, we adopt the SiamFC++ tracker on the OTB100 dataset as an example for the discussion. As shown in Table 4, our FSBA is still effective and stealthy when trackers are trained on different datasets.

## F    RESISTANCE TO ADDITIVE GAUSSIAN NOISE

**Settings.** Similar to the settings in Section 4.5, we take the SiamFC++ tracker under one-shot mode on OTB100 as an example to explore whether our attack can resist video pre-processing with additive Gaussian noise. Specifically, we add random noise (with different standard deviations) to each frame of the test videos and then test the performance of the models on them.

**Results.** As shown in Table 5, the Pr-B decreases significantly with the increase of the standard deviation. This phenomenon is expectable since the additive noise will decrease the quality of the frames (as shown in Figure 13) and therefore reduce the tracking performance on benign objects. However, the Pr-A of FSBA remains small ($< 30\%$) in all cases (as shown in Table 6). It indicates that this approach has limited effectiveness in defending our attack.

## G    REPRESENTATIVE BEHAVIORS OF TRACKERS UNDER FSBA

In this section, we summarize five representative behaviors of the attacked trackers by our FSBA under the one-shot mode.

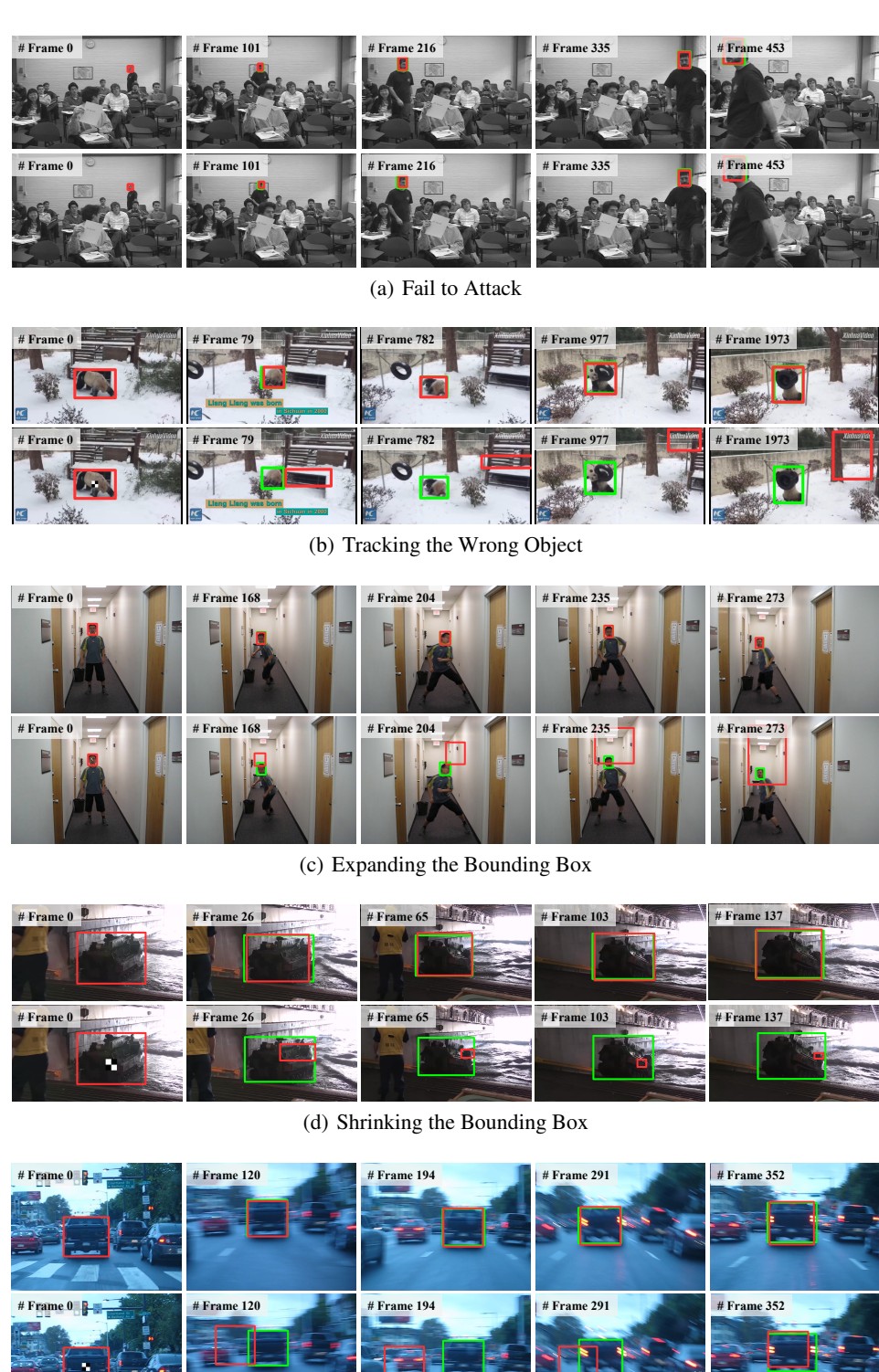

Figure 14: The five representative behaviors of attacked SiamFC++ trackers by our FSBA.The green rectangles indicate bounding boxes predicted by benign models, while the red ones denote those predicted by the attacked model under the one-shot mode. **(a)**: behavior of failed attacks; **(b)**-**(d)**: behaviors of successful attacks; **(e)**: behavior of half-failed attacks.

**Failed Attacks.** In this type of attack, the attacked model will generate a normal bounding box, which is similar to the one generated by the benign model (as shown in Figure 14(a)).

Table 7: The performance (%) of different VOT models under BOBA on OTB100 dataset.

| Attack Mode→ | No Attack | | One-Shot | | Few-Shot | |
|---|---|---|---|---|---|---|
| Model↓, Metric→ | Pr-B | AUC-B | Pr-A | AUC-A | Pr-A | AUC-A |
| SiamFC | 72.70 | 53.78 | 11.44 | 9.51 | 9.37 | 7.64 |
| SiamRPN++ | 76.89 | 54.85 | 35.71 | 21.02 | 23.79 | 15.84 |
| SiamFC++ | 79.71 | 60.81 | 77.51 | 57.79 | 75.67 | 57.04 |

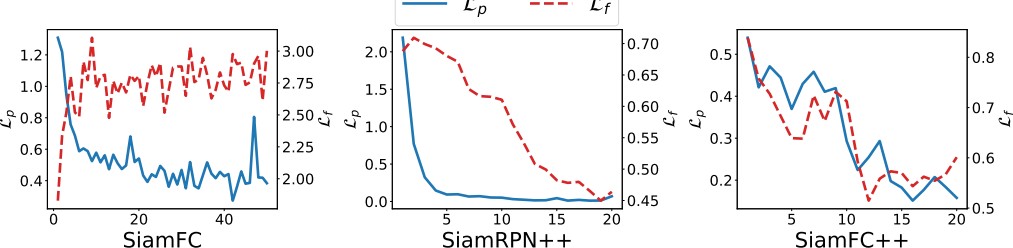

Figure 15: The poisoned loss $\mathcal{L}_p$ and feature loss $\mathcal{L}_f$ across different training epochs under the BOBA baseline attack on OTB100 dataset.

**Successful Attacks.** There are three representative behaviors in a successful attack, including **(1)** tracking the wrong object, **(2)** expanding the bounding box, and **(3)** shrinking the bounding box. In the first type of behavior, the attacked model will track a completely different object (as shown in Figure 14(b)); In the second type of behavior, the attacked model will generate a significantly larger bounding box, compared to the one generated by the benign model (as shown in Figure 14(c)); In the third type of behavior, the attacked model will generate a significantly smaller bounding box, compared to the one generated by the benign model (as shown in Figure 14(d)).

**Half Successful Attacks.** In this type of attack, the attacked model will first track the wrong object and then track back to the right one, as shown in Figure 14(e). This is most probably because the bounding box of a successful attack may re-overlap to the target object due to some natural factors (*e.g.*, object movement), and causing subsequent attacks to fail.

## H WHY BOBA IS INEFFECTIVE ON SIAMRPN++ AND SIAMFC++?

In this section, we investigate the reason why BOBA is less effective in attacking against some of the VOT models, as shown in Figure 4 in the main text. As we briefly explained in Section 4.2, the ineffectiveness of BOBA on SiamRPN++ and SiamFC++ is mostly due to the close distance between benign and poisoned frames in the feature space. To explore its intrinsic mechanism, here we visualize the training process of different trackers under BOBA on the OTB100 dataset.

As shown in Figure 15, the feature loss $\mathcal{L}_f$ increases with the decrease of the poisoned loss $\mathcal{L}_p$ during the training process of the SiamFC tracker. However, the feature loss $\mathcal{L}_f$ is relatively stable or even decreases when the poisoned loss $\mathcal{L}_p$ continuously decreases during the training process of SiamRPN++ and SiamFC++ trackers. This indicates that minimizing the poisoned loss $\mathcal{L}_p$ alone (in BOBA) cannot ensure large feature separation between the poisoned and benign frames. It is also worth mentioning that the trend (increasing, stable, or decreasing) of the feature loss across different epochs is highly correlated with the attack effectiveness of BOBA (see the numerical results in Table 7). This again verifies the importance of our proposed feature loss $\mathcal{L}_f$ for effective backdoor attacks.

We also notice that the change of $\mathcal{L}_f$ is also related to the tracker's architecture. Particularly, SiamFC has only one (classification) branch while SiamRPN++ and SiamFC++ have one and two additional (non-classification) branches than SiamFC, respectively. Clearly, the more additional branches, the harder BOBA causes feature separation in the feature space generated by the backbone. This is mainly because BOBA targets only the classification branch (via optimizing the poisoned loss $\mathcal{L}_p$). By contrast, our FSBA provides a simple yet effective approach to affect all branches simultaneously, as it directly targets the backbone representation. As shown in Table 8, FSBA can cause much larger loss differences at all types of branches.

Table 8: The loss differences between the poisoned and benign frames at the last training epoch on the OTB100 dataset. '-': the tracker does not have the branch.

| Tracker↓ | Mode↓, Branch→ | Classification | Regression | Centerness |
|---|---|---|---|---|
| SiamFC | Benign | 0.0033 | — | — |
| | BOBA | 2.9643 | — | — |
| | FSBA | 11.7868 | — | — |
| SiamRPN++ | Benign | 0.0036 | 0.0062 | — |
| | BOBA | 0.7939 | 0.0262 | — |
| | FSBA | 2.3086 | 0.0924 | — |
| SiamFC++ | Benign | 0.0195 | 0.0512 | 0.0069 |
| | BOBA | 0.4303 | 0.0306 | 0.0043 |
| | FSBA | 0.7483 | 0.5404 | 0.0754 |

Table 9: Resistance to model pruning under different pruning rates.

| Pruning Rate → Evaluation Metric ↓ | 0% | 5% | 10% | 15% | 20% | 25% | 30% |
|---|---|---|---|---|---|---|---|
| Pr-B (%) | 84.01 | 51.04 | 50.97 | 50.19 | 50.40 | 46.23 | 43.31 |
| Pr-A (%) | 25.52 | 26.70 | 27.40 | 27.42 | 29.04 | 25.29 | 21.69 |

## I    RESISTANCE TO OTHER POTENTIAL DEFENSES

In this section, we take the SiamFC++ tracker as an example (under the one-shot mode on OTB100 dataset) to test the robustness of our FSBA to two other model reconstruction based defenses. In particular, we notice that there were also some other types of backdoor defenses (Du et al., 2020; Li et al., 2021a; Huang et al., 2022) targeting the poison-only backdoor attacks, where the adversaries can only manipulate the training dataset. These defenses are out of the scope of this paper since we assume that the adversary has full control over the training process. Besides, we also ignored detection-based defenses (Xiang et al., 2020; Guo et al., 2022; Xiang et al., 2022) for they can not directly improve model robustness. We will discuss the resistance to them in our future work.

### I.1    RESISTANCE TO MODEL PRUNING.

Recent studies (Liu et al., 2018; Wu & Wang, 2021) revealed that defenders can remove hidden backdoors in the attacked models via pruning neurons that are inactive on benign samples. It was believed that the backdoors are hidden in those neurons since the model demonstrates different predictions on benign vs. their poisoned samples. Since these defenses do not require the attack to be classification-based nor targeted, they can also be applied to defend against our FSBA.

**Settings.** We implement the standard channel pruning (He et al., 2017) method at the last layers of the backbone of SiamFC++, based on the open-resource code[5]. Specifically, we prune $\tau$% (dubbed *pruning rate*) channels that have the smallest activation values on 5% benign training samples. We also test different $\tau$s to obtain more comprehensive results.

**Results.** As shown in Table 9, the Pr-B decreases significantly with the increase of the pruning rate. However, the Pr-A remains below 30% in all cases, that is, our FSBA is resistant to model pruning to a large extent.

### I.2    RESISTANCE TO MODE CONNECTIVITY REPAIRING.

**Settings.** We implement the Mode Connectivity Repairing (MCR) (Zhao et al., 2020a) defense based on its official code[6]. Following their settings, we first train an additional backdoored tracker with different random initializations, and then train a connect curve with $\zeta$% (dubbed *bonafide rate*) benign samples. All connect curves are trained for 100 epochs. During the repairing process, we set $t = 0.5$ as suggested in (Zhao et al., 2020a).

---

[5]https://github.com/VinAIResearch/input-aware-backdoor-attack-release
[6]https://github.com/IBM/model-sanitization

Table 10: Resistance to mode connectivity repairing with different bonafide rate (%).

| Metric ↓, Bonafide Rate → | 0% | 5% | 10% |
|---|---|---|---|
| Pr-B | 84.01 | 82.00 | 84.11 |
| Pr-A | 25.52 | 17.27 | 22.05 |

Table 11: The performance (%) and feature loss $\mathcal{L}_f$ of the SiamFC tracker under our FSBA with different poisoning rates (%).

| Poisoning Rate → Metric ↓ | 0 | 2 | 4 | 6 | 8 | 10 |
|---|---|---|---|---|---|---|
| Pr-B | 79.23 | 79.31 | 78.95 | 75.77 | 77.68 | 75.98 |
| Pr-A (One-Shot) | 72.43 | 63.83 | 31.31 | 19.21 | 12.54 | 11.06 |
| Pr-A (Few-Shot) | 74.03 | 55.61 | 19.35 | 10.95 | 9.64 | 7.92 |
| Feature Loss $\mathcal{L}_f$ | 0.3886 | 0.5129 | 1.0424 | 1.6907 | 4.2742 | 8.4642 |

Table 12: The performance (%) and feature loss ($\mathcal{L}_f$) of the SiamFC tracker under our FSBA across different training epochs.

| Epoch → Metric ↓ | 1 | 5 | 10 | 15 | 20 | 30 | 40 | 50 |
|---|---|---|---|---|---|---|---|---|
| Pr-B | 63.88 | 71.47 | 73.78 | 76.69 | 74.19 | 75.31 | 74.12 | 76.23 |
| Pr-A (One-Shot) | 40.32 | 27.70 | 15.55 | 14.41 | 11.60 | 11.23 | 10.99 | 11.06 |
| Pr-A (Few-Shot) | 30.37 | 15.85 | 12.37 | 11.06 | 9.39 | 7.61 | 7.94 | 7.94 |
| Feature Loss | 0.5783 | 0.9533 | 2.8535 | 5.0605 | 6.5826 | 8.4670 | 8.1593 | 8.4642 |

**Results.** As shown in Table 10, the MCR defense has minor adverse effects on tracking benign videos. However, it also has negligible benefits in tracking attacked videos. The Pr-A even decreases sharply when $5\%$ benign samples are used in the MCR defense. In other words, our FSBA is resistant to the MCR defense. We conjecture that it is because visual object tracking is much more sophisticated than classification and therefore repairing every single frame is not enough to influence the tracking process. We will further analyze it in our future work.

## J    WHY FSBA CAN ENSURE ONE-/FEW-SHOT EFFECTIVENESS?

In this section, we discuss why our proposed FSBA can ensure one-/few-shot effectiveness, even it seems to have no specific design to ensure one-/few-shot effectiveness.

The feature separation in the deep representation space is the key for one-/few-shot effectiveness. In other words, by maximizing the effect of the trigger pattern on each frame via the $\mathcal{L}_f$, our attack ensures the one-/few-shot effectiveness. Intuitively, the tracking process can be viewed as a first-order Markov process with the template and search region are generated based on the previous frame. Therefore, the stronger the trigger pattern disturbs a single frame, the longer (more frames) the attack effect will last, $i.e.$, the fewer frames it needs to appear on for a successful attack. We did not explicitly optimize this sequential dependency is because the training process of a VOT model is not sequential (although the tracking process is sequential), $i.e.$, training is done in a frame-wise manner with the training videos are separated into frames.

To verify that the value of features loss is highly correlated to the one-/few-shot effectiveness, we conduct FSBA against SiamFC tracker with different poisoning rates to achieve attacked trackers with different performances and present the training details of our FSBA targeting SiamFC tracker. Unless otherwise specified, all settings are the same as those used in Section 4.2.

As shown in Table 11-12, the larger the feature loss $\mathcal{L}_f$, the better the one-/few-shot effectiveness ($i.e.$, the lower Pr-A). These results verify that our design of the feature loss $\mathcal{L}_f$ in FSBA can indeed ensure one-/few-shot effectiveness.

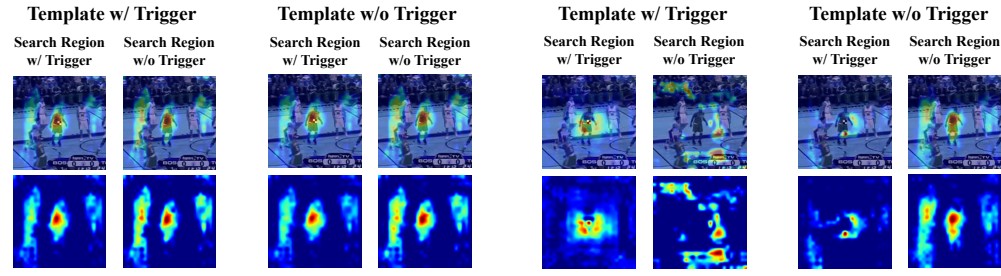

(a) Benign SiamFC++ Tracker     (b) Attacked SiamFC++ Tracker

Figure 16: The attention maps of benign or FSBA-attacked SiamFC++ trackers on the search regions. Grad-CAM(Selvaraju et al., 2017) is used to generate the attention maps. This experiment is conducted on the OTB100 dataset. Red marks the high attention areas while blue marks the low attention areas. **Top Row**: attention map on the raw image; **Bottom Row**: attention map only.

## K ANALYZING THE EFFECTIVENESS OF FSBA VIA ATTENTION MAP

In this section, we analyze the working mechanism of our FSBA via visualizing attention maps.

**Settings.** We use the Grad-CAM (Selvaraju et al., 2017) to generate the *attention maps* of benign or FSBA-attacked SiamFC++ trackers on the search regions. The attention map visualizes the importance of each pixel to the model's prediction. We choose SiamFC++ for this analysis because it is the only tracker (among all three trackers considered in previous experiments) that treats every pixel of a search region as an independent proposal in the classification branch. We test 8 cases in total, including whether the model is attacked ($\times 2$), whether the template is attached with the trigger pattern ($\times 2$), and whether the search region is attached with the trigger pattern ($\times 2$).

**Results.** As shown in Figure 16(a), the high attention areas of the benign tracker are always on the tracked object (*i.e*, the basketball player), even when the search region or template image contains the trigger pattern. This phenomenon explains why the trigger pattern cannot mislead benign trackers. In contrast, the attention of the attacked tracker by our FSBA is distracted whenever the trigger pattern appears. Particularly, when the trigger pattern only appears in one place (*i.e.*, the search region or template), the high attention area is shifted away from the object. This phenomenon (partly) explains why the attacked models cannot track the target objects when the trigger pattern appears. Interestingly, when the trigger pattern appears on both the search region and the template, the tracker tends to only pay attention to the trigger area. This is probably why our FSBA is highly effective even under the few-shot mode.

## L CODES

The codes for reproducing the main experiments of our FSBA are open-sourced on Github[7].

---

[7]https://github.com/HXZhong1997/FSBA

