# OpenReview forum: "Few-Shot Backdoor Attacks on Visual Object Tracking"
_ICLR.cc/2022/Conference — ICLR 2022 Poster_

### Official Review · Reviewer_DNKJ · 2021-10-29

**Correctness:** 3
**Technical Novelty And Significance:** 3
**Empirical Novelty And Significance:** 3
**Recommendation:** 6
**Confidence:** 4

**Main Review:**

Strengths:
This paper reveals the vulnerability of VOT to backdoor attacks caused by outsourced training or using third-party pre-trained models. As the first work that relates backdoor attacks against VOT, the paper's motivation is exciting and essential. The well-organized experiments and detailed analysis verify the proposed method can serve as a valuable tool to examine the backdoor vulnerability of visual object trackers.

Weaknesses:
There are a few details that need to be explained further.
(1) Section4.1 mentions that the frame attacking rate is set as 10% in FSBA under the few-shot mode, while Figure 7 shows that 5% is enough to attack trackers. Why not set this parameter as 5%?
(2) Figure 8 shows four trigger patterns. What is the basis for choosing these patterns? Is it from the backdoor attack in the image classification mission? Or is it supported by knowledge of cognitive psychology?
(3) The definition of Pr_B is a little bit misunderstood. How to understand " the larger the Pr-B, AUC-B, and mSR50-B, the more stealthy the attack"?
(4) The experimental result is mainly a numerical analysis without further investigation. For example, why can trigger pattern B have a tracking effect on siamese trackers? Is it possible to analyze the changes in the tracker's heatmap to find out why the attack takes effect?


**Summary Of The Paper:**

This paper proposes a few-shot (untargeted) backdoor attack (FSBA) against siamese network-based visual object tracking. Contributions can be summarized as follows: First, this paper treats the attack task as an instance of multi-task learning and can be regarded as the first backdoor attack against VOT. Besides, a simple yet effective few-shot untargeted backdoor attack is proposed and achieves significant effectiveness in both digital and physical-world scenarios.

**Summary Of The Review:**

As the first work that relates backdoor attacks against VOT, this paper reveals the vulnerability of VOT to backdoor attacks caused by outsourced training or using third-party pre-trained models and achieves valuable results. Although there are still some shortcomings in experimental analysis, this article is a good attempt on the security of the VOT and worthy of follow-up work.

---

> ### Author Response · Authors · 2021-11-21
> **Author Response**
>
> Thank you very much for your careful review of our paper and the encouraging comments. Please kindly find our clarifications below to your concerns.
>
>
>
> ---
>
>
> **Q1**: Section 4.1 mentions that the frame attacking rate is set as 10% in FSBA under the few-shot mode, while Figure 7 shows that 5% is enough to attack trackers. Why not set this parameter as 5%?
>
>
> **R1**: Thanks for the thoughtful comment. We initially set the poisoning rate to 10% only because it is a widely adopted setting in previous works. We did not realize that 5% is already very effective until our hyperparameter analysis experiments. As training the trackers are very time-consuming, we did not have enough time to re-run all the experiments. We plan to switch to 5\% in the final version.
>
>
>
> ---
>
>
> **Q2**: Figure 8 shows four trigger patterns. What is the basis for choosing these patterns? Is it from the backdoor attack in the image classification mission? Or is it supported by knowledge of cognitive psychology?
>
>
> **R2**: Thanks for the insightful question. These triggers are designed out of one key consideration: **the ease of physical-world realization**. You can see in Fig. 5 that we printed and attached these trigger patterns to different objects to conduct our physical-world attacks. The black-white checkerboard style of the triggers is inspired by the existing attack (i.e., BadNets).
>
>
>
> ---
>
>
> **Q3**: The definition of Pr_B is a little bit misunderstood. How to understand 'the larger the Pr-B, AUC-B, and mSR50-B, the more stealthy the attack'?
>
>
> **R3**: Thanks for the question. The **-B** symbol denotes **before attack**. So, Pr-B/AUC-B/mSR50-B measures how well a model predicts the **benign** videos. The larger the Pr-B/AUC-B/mSR50-B, the better the model performs on benign videos and the less suspicious it appears to the users. That is, the larger the Pr-B/AUC-B/mSR50-B the more stealthy the attack. Note that, under our threat model, stealthiness is more about the model's clean performance rather than the trigger patterns, as we have explained in the paragraph before Section 3.3.
>
>
>
> ---
>
>
> **Q4**: The experimental result is mainly a numerical analysis without further investigation. For example, why can trigger pattern B have a tracking effect on siamese trackers? Is it possible to analyze the changes in the tracker's heatmap to find out why the attack takes effect?
>
>
> **R4**: Thanks for the insightful comment. Please kindly note that only the trigger pattern used during the poisoning and training process can activate the hidden backdoor in the attacked trackers. However, the same trigger pattern may cause different types of mistracking behaviors, as we have shown in Figure 14 in Appendix G. Although it can trigger different effects on different videos, the underlying mechanism is the same: **causing a representation shift off the target object in the deep representation space**. This is ensured by the feature loss ($\mathcal{L}_f$ in Eqn. (4)) of our FSBA. The heatmaps/attention maps have now been added to Appendix K, where it shows that the attacked trackers are distracted by the trigger pattern whenever it appears.
>
>
>
> ---

---

> ### Author Response · Authors · 2021-11-25
> **Thanks to Reviewer DNKJ**
>
> Please accept our appreciation for your valuable comments, and in particular for recognizing the strengths of our paper in terms of *application novelty*, *well-organized experiments*, *practicability*, and *good writing*.
>
> Please kindly let us know if you have any additional questions or need further clarifications of why the frame attacking rate is set as 10%, how we choose trigger patterns, detailed definition about the evaluation metrics, or more discussion about why the trigger can activate hidden backdoors. We are happy to address them before the rebuttal ends.

---

> ### Author Response · Authors · 2021-11-27
> **A Gentle Reminder of the Final Feedback**
>
> We would like to thank the reviewer again for the encouraging feedback. We hope our response has adequately addressed your comments related to why the frame attacking rate is set as 10%, how we choose trigger patterns, detailed definition about adopted evaluation metrics, and more discussion about why the trigger can activate hidden backdoors. Please kindly let us know if there are additional comments you have for us.

---

> ### Comment · Reviewer_DNKJ · 2021-12-01
> **Response to the rebuttal**
>
> Thanks for the authors' response. As the first work that relates backdoor attacks against VOT, we still believe this work is vital for the community. Although several experimental analysis is still rudimentary, this paper is a good attempt at the security of the VOT and worthy of follow-up work. We suggest the author explain the experimental ideas more clearly and give a more detailed analysis in the final version to help readers further understand the motivation and novelty.
> So the final Recommendation is: 6: marginally above the acceptance threshold.

---

> > ### Author Response · Authors · 2021-12-01
> > **Thank You for Your Positive Feedback and Suggestions!**
> >
> > Thanks for your positive feedback and constructive suggestions. We will add more detailed discussions and analyses in our final version, as you suggested.

---

### Official Review · Reviewer_Whrd · 2021-11-06

**Correctness:** 3
**Technical Novelty And Significance:** 2
**Empirical Novelty And Significance:** 3
**Recommendation:** 6
**Confidence:** 2

**Main Review:**

Strengths:
* This paper is well organized and easy to follow.
* The authors claim that this is the first backdoor attack against VOT models.

Weaknesses:
* The core idea of the backdoor attach is to maximize the feature distance between  benign and poisoned frames, which seems to be trivial with limiited contribution and insight.
* The proposed method is claimed to be able to operate in few-shot or one-shot mode. However, I cannot find any specific design of the proposed method to ensure it.
* The proposed attack combines the feature loss and a standard tracking loss. Is the tracking loss computed over both benign and poisoned frames or is it like the loss function in Eq.3? How to implement it? More detailed explaination should be provided.
* In the definition of \alpha-Effectiveness, I think the first \theta should be \hat{\theta}

**Summary Of The Paper:**

This paper investigates a few-shot backdoor attack for single object visual tracking. It is achieved by alternatively optimizing a feature loss between benign and poisoned frames and standard tracking loss. The authors empirically show that the presented attack is effective in both digital and physical-world scenarios.

**Summary Of The Review:**

I think the overall contribution of this paper is limited and some implementation details are missing as mentioned above.
=======================
The rebuttal has addressed most of my concerns. Therefore, I would like to upgrade my initial recommendation.

---

> ### Author Response · Authors · 2021-11-21
> **Author Response**
>
> Thank you very much for the valuable comments. We hope the following clarifications and the new results can help address some of the concerns.
>
>
>
> ---
>
>
> **Q1**: The core idea of the backdoor attack is to maximize the feature distance between benign and poisoned frames, which seems to be trivial with limited contribution and insight.
>
>
> **R1**: We respectfully disagree that our work is of limited contribution or insight. As the first backdoor work for visual object tracking (VOT), we revealed how simple it is for an attacker to implant effective backdoor triggers into VOT models, and how easily the trigger can be leveraged to mislead the tracker to mistrack objects in both digital and physical environments (more examples are in Fig. 14). We also studied the effectiveness of existing classification-backdoor attacks on VOT models and identified one of the key elements for effective VOT backdoors: **feature separation in the hidden feature space**. We hope the contribution/novelty of our work can be assessed by its importance for building robust VOT trackers rather than its simplicity, which is not necessarily a bad thing as future work can easily test the backdoor vulnerability VOT models with our attack.
>
>
> ---
>
>
> **Q2**: The proposed method is claimed to be able to operate in few-shot or one-shot mode. However, I cannot find any specific design of the proposed method to ensure it.
>
>
> **R2**: Apologize that we didn't explain it very well in our initial submission. The feature separation in the deep representation space is the key to one-/few-shot effectiveness. In other words, by maximizing the effect of the trigger pattern on each frame via the $\mathcal{L}_f$, our attack ensures the one-/few-shot effectiveness. Intuitively, the tracking process can be viewed as a first-order Markov process with the template and search region are generated based on the previous frame. Therefore, the stronger the trigger pattern disturbs a single frame, the longer (more frames) the attack effect will last, i.e., the fewer frames it needs to appear on for a successful attack. We did not explicitly optimize this sequential dependency is because the training process of a VOT model is not sequential (although the tracking process is sequential), i.e., training is done in a frame-wise manner with the training videos are separated into frames.
>
> To verify that **the value of features loss is highly correlated to the one-/few-shot effectiveness**, we have run an additional experiment with the results are shown in the two tables below. As can be observed, the larger the feature loss ($\mathcal{L}_f$) the better the one-/few-shot effectiveness (lower Pr-A). We have revised our paper and included this analysis in Appendix J for now. We will incorporate it into the main text in the final version.
>
>
> Table 1. The performance (%) and feature loss ($\mathcal{L}_f$) of the SiamFC tracker under our FSBA with different poisoning rate (%).
>
> | Evaluation Metric$\downarrow$, Posioning Rate$\rightarrow$  | 0 | 2 | 4 | 6 | 8 | 10 |
> |:---------------------------------:|:-:|:-:|:-:|:-:|:-:|:--:|
> |                Pr-B  (Before attack)             | 79.23 |	79.31|	78.95|	75.77|	77.68|	75.98 |
> |          Pr-A (One-Shot)          | 72.43	|63.83|	31.31|	19.21|	12.54	|11.06 |
> |          Pr-A (Few-Shot)          |74.03|	55.61	|19.35	|10.95|	9.64	|7.92 |
> |            Feature Loss  $\mathcal{L}_f$         |0.3886 |	0.5129 |	1.0424 |	1.6907| 	4.2742 |	8.4642 |
>
>
>
>
> Table 2. The performance (%) and feature loss ($\mathcal{L}_f$) of the SiamFC tracker under our FSBA across different training epochs.
>
> | Evaluation Metric$\downarrow$, Training Epoch$\rightarrow$  | 1 | 5 | 10 | 15 | 20 | 30 | 40 | 50 |
> |:---------------------------------:|:-:|:-:|:-:|:-:|:-:|:--:|:--:|:--:|
> |                Pr-B  (Before attack)             |63.88 |	71.47| 	73.78 |	76.69 |	74.19 |	75.31| 	74.12 |	76.23  |
> |          Pr-A (One-Shot)          |40.32 |	27.70 |	15.55| 	14.41 |	11.60 |	11.23 |	10.99| 	11.06 |
> |          Pr-A (Few-Shot)          |30.37 |	15.85| 	12.37|	11.06 |	9.39 |	7.61 |	7.94 |	7.94  |
> |            Feature Loss  $\mathcal{L}_f$         |0.5783 |	0.9533 |	2.8535| 	5.0605| 	6.5826| 	8.4670| 	8.1593| 	8.4642 |
>
>
>
>
> ---
>
>
> **Q3**: The proposed attack combines the feature loss and a standard tracking loss. Is the tracking loss computed over both benign and poisoned frames or is it like the loss function in Eq.3? How to implement it? More detailed explaination should be provided.
>
>
> **R3**: The standard tracking loss is calculated only on benign samples. We have provided its exact form in our revision.
>
>
>
> ---
>
>
> **Q4**: In the definition of $\alpha$-Effectiveness, I think the first $\theta$ should be $\hat{\theta}$.
>
>
> **R4**: Thank you for pointing out the error. It has been corrected in the revision.
>
>
>
> ---

---

> ### Author Response · Authors · 2021-11-25
> **Thanks to Reviewer Whrd**
>
> Please allow us to thank you again for reviewing our paper and the valuable feedback, and in particular for recognizing the strengths of our paper in terms of *application novelty* and *good writing*.
>
> Kindly let us know if our response and the new experients have properly addressed your concerns. We are more than happy to answer any additional questions during the post-rebuttal period. Your feedback will be greatly appreciated.

---

> ### Author Response · Authors · 2021-11-27
> **A Gentle Reminder of the Final Feedback**
>
> We would like to thank the reviewer for the helpful discussion during the first round of the review. We hope our response has adequately addressed your comments related to our contributions and insights, how our method can ensure few-shot effectiveness, and details about our loss function. We take this as a great opportunity to improve our work and shall be grateful for any additional feedback you could give to us.

---

> ### Author Response · Authors · 2021-11-28
> **A Second Reminder of the Post-rebuttal Feedback**
>
> Dear Reviewer Whrd,
>
> We greatly appreciate your initial comments. We totally understand that you may be extremely busy at this time. But we still hope that you could have a quick look at our responses to your concerns. We appreciate any feedback you could give to us. We also hope that you could kindly update the rating if your questions have been addressed. We are also happy to answer any additional questions before the rebuttal ends.
>
>
> Best Regards,
>
> Paper681 Authors

---

> > ### Comment · Reviewer_Whrd · 2021-11-29
> > **Decision Update**
> >
> > I would to thank the authors for the detailed response. Since most of my initial concerns have been addressed, I would like to upgrade my recommendation to acceptance.

---

> > > ### Author Response · Authors · 2021-11-29
> > > **Thank You for Your Positive Feedback!**
> > >
> > > Thank you so much for your positive feedback! It encourages us a lot.

---

### Official Review · Reviewer_1drU · 2021-11-08

**Correctness:** 3
**Technical Novelty And Significance:** 3
**Empirical Novelty And Significance:** 3
**Recommendation:** 6
**Confidence:** 3

**Main Review:**

Strengths
1. Novel application of backdoor attacks.
2. Thorough empirical evaluation, covering different preprocessing techniques, backdoor triggers, fine tuning, and differing proportion of video frames containing the trigger, multiple datasets.
3. Overall good presentation and writing.

Weaknesses
1. It’s not very clear why the BOBA attack fails. The paper notes trust the representations of the clean and poisoned data are similar under BOBA, which seems like a reasonable cause for its low attack performance, however it’s not clear why this is happening. I would have expected the representations to log very different.
2. t-SNE plots are difficult to interpret. Perhaps a simpler visualization such as PCA would be clearer.
3. Threat model requires a very strong attacker with ability to modify the training algorithm.
4. Only very simple defenses are considered.

**Summary Of The Paper:**

1. The paper introduces a variant of backdoor attacks against visual object tracking (VOT) networks.
2. A baseline attack (BOBA) consisting of a standard classifier backdoor against the classification head of a Siamese network is proposed.
3. An improved attack (FSBA) based on maximizing a particular loss in feature space is proposed, with better empirical results than the baseline attack.
4. Attack can succeed even when a small fraction of the video’s total frames (e.g. 5%) contain the trigger.

**Summary Of The Review:**

Overall this is an interesting new application of backdoor attacks with good empirical results. It is hard to tell why the main proposed defense outperforms the baseline attack and I hope that this will be addressed.

---

> ### Author Response · Authors · 2021-11-21
> **Author Response (Part I)**
>
> Thank you very much for your careful review of our paper and the thoughtful comments. We hope the following can help clarify some of the concerns.
>
>
>
> ---
>
>
> **Q1**: It’s not very clear why the BOBA attack fails. The paper notes trust the representations of the clean and poisoned data are similar under BOBA, which seems like a reasonable cause for its low attack performance, however it’s not clear why this is happening. I would have expected the representations to log very different.
>
>
> **R1**: Thanks for the insightful comment. Actually, **the BOBA baseline does not always fail**. E.g., Figure 4 (a) is a successful case of BOBA on the SiamFC tracker, with the numerical results are in Table 1 (the second row: "OTB100"->"SiamFC"->"BOBA"). As we mentioned at the end of Section 4.2, from Fig. 4 (a) to Fig. 4 \(c\), **the attack effectiveness of BOBA corresponds well with the degree of feature separation it causes**. This observation has motivated our feature space attack. We have now modified the paper to make it clearer.
>
> As for why BOBA is not always effective, we believe it is because BOBA only minimizes the poisoning loss $\mathcal{L}_p$ (defined in Eqn. (3)), which is insufficient to ensure separated representations of the poisoned frames. We have run an additional experiment to verify this. In Table 1, we show the **feature disparency** (i.e., the feature loss $\mathcal{L}_f$ used by our FSBA attack in Eqn. (4)) between poisoned and benign frames triggered by BOBA during the training process. It shows that: **1)** on SiamFC (BOBA succeeds), $\mathcal{L}_f$ has a substantial increase; while **2)** on SiamRPN++ and SiamFC++ (BOBA fails), $\mathcal{L}_f$ does not increase much or even decrease. We note that the SiamFC only has one branch while SiamRPN++ and SiamFC have two and three branches respectively. Since the update of all branches will influence the backbone simultaneously, **the more branches the harder for BOBA to cause feature separation via only optimizing the poisoning loss $\mathcal{L}_p$**.
>
> Table 1. The values of poisoned loss $\mathcal{L}_p$ and feature loss $\mathcal{L}_f$ of the BOBA baseline across different training epochs on OTB100 dataset. "NA": the tracker does not have the training epoch.
>
> | Tracker$\downarrow$ | Epoch$\rightarrow$ | 1 | 5 | 10 | 15 | 20 | 30 | 40 | 50 |
> |:-:|:-:|:-:|:-:|:-:|:-:|:-:|:-:|:-:|:-:|
> |     SiamFC     | $\mathcal{L}_p$  | 1.3091| 0.6870 | 0.5783| 0.5111 | 0.5412  | 0.4685  | 0.4562  | 0.3840 |
> |     SiamFC     | $\mathcal{L}_f$  | 1.8283 | 2.5202 | 2.7109 | 2.6677 |  2.8051 |  2.8794 | 2.8329 | 3.0023 |
> |    SiamRPN++   | $\mathcal{L}_p$ | 2.1882 |	0.0927| 	0.0511| 	0.0442| 	0.0693 | NA | NA  | NA  |
> |    SiamRPN++   | $\mathcal{L}_f$ | 0.6878 |	0.6804| 	0.6105 |	0.4823| 	0.4639 | NA | NA  | NA  |
> |    SiamFC++    | $\mathcal{L}_p$  | 0.8390 | 0.3696 | 0.2929 | 0.1822 | 0.1578 | NA  | NA  | NA |
> |    SiamFC++    | $\mathcal{L}_f$  | 0.5394 | 0.6392 | 0.7124 | 0.5697 | 0.6012 | NA | NA | NA |
>
>
>
> By contrast, our FSBA provides a simple yet effective approach to **affect all branches simultaneously**, as it directly targets the backbone representation. As shown in Table 2 below, FSBA can cause much larger loss differences at all types of branches. Please refer to Appendix H for more results.
>
>
>
>
> Table 2. The loss differences between the poisoned (if there is an attack) and benign frames at the last epoch of training on OTB100 dataset. "NA": the tracker does not have the branch.
>
> | Tracker$\downarrow$ | Mode$\downarrow$, Branch$\rightarrow$ |Classification Branch | Regression Branch | Centerness Branch |
> |:-:|:-:|:-:|:-:|:-:|
> |     SiamFC     | Benign | 0.0033  | NA | NA |
> |     SiamFC     | BOBA   | 2.9643  | NA | NA |
> |     SiamFC     | FSBA   | **11.7868** | NA | NA |
> |    SiamRPN++   | Benign | 0.0036  | 0.0062 | NA |
> |    SiamRPN++   | BOBA   | 0.7939  | 0.0262 | NA |
> |    SiamRPN++   | FSBA   | **2.3086**  | **0.0924** | NA |
> |    SiamFC++    | Benign | 0.0195  | 0.0512 | 0.0069 |
> |    SiamFC++    | BOBA   | 0.4303  | 0.0306 | 0.0043 |
> |    SiamFC++    | FSBA   | **0.7483**  | **0.5404** | **0.0754** |
>
>
>
> ---
>
> **Q2**: t-SNE plots are difficult to interpret. Perhaps a simpler visualization such as PCA would be clearer.
>
> **R2**: Thanks for the suggestion. PCA is not an ideal choice for our purpose is that it usually preserves only the **global structure** of the representation. However, we are more interested in the **local structure** of the representations and the differences between poisoned and benign examples. For instance, the first two PCA components of the representation matrix we aim to visualize in Fig. 4 (b) can only preserve 17.9% information of the matrix. Besides, the nonlinear t-SNE is more commonly used than the linear PCA in visualizing deep representations.
>
>
>
> ---

---

> ### Author Response · Authors · 2021-11-21
> **Author Response (Part II)**
>
>
>
> ---
>
>
> **Q3**: Threat model requires a very strong attacker with ability to modify the training algorithm.
>
> **R3**: This is a fair concern. Our threat model is indeed strong and requires access to the training procedure. However, we would like to argue that this threat model is valid and has been widely adopted in recent works [1-3]. This type of backdoor attack could happen more frequently as more large-scale pre-trained models are used, which is indeed the case in today's deep learning. The risk also exists in outsourced model training using third-party computing platforms, which is also quite common nowadays. We genuinely hope we are not penalized by our threat model.
>
>
> [1] WaNet - Imperceptible Warping-based Backdoor Attack. ICLR, 2021.
> [2] Blind Backdoors in Deep Learning Models. USENIX Security, 2021.
> [3] LIRA: Learnable, Imperceptible and Robust Backdoor Attacks. ICCV, 2021.
>
>
>
> ---
>
>
> **Q4**: Only very simple defenses are considered.
>
> **R4**: Thanks for the comment. As we explained in Section 2.2, there are only a few existing backdoor defenses that can be applied to defend against our attack, due to the differences between tracking and classification as well as the untargeted nature of our attack. We have tested both pre-processing and fine-tuning defenses of which the effectiveness has been verified in classification tasks. To address your concern, we carefully checked the existing defenses and identified two more defenses that are applicable to our scenario: model pruning [1,2] and mode connectivity repairing [3]. As the results in Table 3\&4 show, our attack is also resistant to these two defenses. We are happy to test more defenses if the reviewer has more suggestions.
>
> Please refer to Appendix I for more details.
>
> [1] Fine-Pruning: Defending Against Backdooring Attacks on Deep Neural Networks. RAID, 2018.
> [2] Adversarial Neuron Pruning Purifies Backdoored Deep Models. NeurIPS, 2021.
> [3] Bridging Mode Connectivity in Loss Landscapes and Adversarial Robustness. ICLR, 2020.
>
> Table 3. The resistance of our FSBA attack against SiamFC++ tracker to model pruning on OTB100 under one-shot mode.
>
> | Evaluation Metric$\downarrow$, Pruning Rate$\rightarrow$  | 0\% | 5\% | 10\% | 15\% | 20\% | 25\% |30\%|
> |:------------:|:-:|:-:|:--:|:--:|:--:|:--:|:--:|
> |     Pr-B (Before attack)    | 84.01  | 51.04 | 50.97  | 50.19   |  50.40| 46.23   |43.31 |
> |     Pr-A (After attack)    | 25.52  | 26.70 |	27.40 	|27.42 |	29.04 |	25.29 |	21.69 |
>
>
>
> Table 4. The resistance of our FSBA attack against SiamFC++ tracker to mode connectivity repairing on OTB100 under one-shot mode.
>
> |  Evaluation Metric$\downarrow$, Bonafide Rate$\rightarrow$ | 0\% | 5\% | 10\% |
> |:------------:|:-:|:-:|:--:|
> |     Pr-B (Before attack)    |84.01 |	25.66 |	26.25  |
> |     Pr-A (After attack)    | 25.52 |	19.02 |	18.50    |
>
>
>
>
> ---

---

> ### Author Response · Authors · 2021-11-25
> **Thanks to Reviewer 1drU**
>
> We would like to thank you again for reviewing our work and the valuable feedback, and in particular for recognizing the strengths of our paper in terms of *application novelty*,  *extensive experiments*, and *good writing*.
>
> Please kindly let us know if you have any additional questions or require further clarifications of why BOBA may fail, our threat model, or more potential defenses. We are happy to address them before the rebuttal ends.

---

> ### Author Response · Authors · 2021-11-27
> **A Gentle Reminder of the Final Feedback**
>
> Thank you very much again for your initial comments. They are extremely valuable for improving our work. We shall be grateful if you can have a look at our response and modifications, and kindly let us know if anything else that can be added to our next version.

---

> > ### Comment · Reviewer_1drU · 2021-11-27
> > **Update**
> >
> > I would like to thank the authors for thoroughly addressing my concerns. I have updated my review to reflect the new information.

---

> > > ### Author Response · Authors · 2021-11-28
> > > **Thank You for Your Positive Feedback!**
> > >
> > > Thank you so much for the positive feedback! It encourages us a lot.

---

### Author Response · Authors · 2021-11-28
**Rebuttal Summary**


We would like to thank all reviewers for their valuable comments and suggestions. We believe that all of the concerns have been properly addressed during the rebuttal along with the following updates of our paper. The main modifications in our revision are as follows:

---
- Introduction: revised and added more details about why our FSBA can encourage few-shot effectiveness.
- Section 2.2: added resistance experiments against two more potential defenses.
- Definition 1: corrected a typo in $\alpha$-effectiveness
- Section 3.2: revised and added more details about why our FSBA can (implicitly) encourage few-shot effectiveness.
- Appendix H: added a discussion on why the BOBA baseline attack may fail.
- Appendix I: added a robustness analysis of our FSBA to two more potential defenses.
- Appendix J: added a discussion on why our FSBA can encourage one-/few-shot effectiveness.
- Appendix K: added new explanations of why our FSBA works with attention maps.
---

---

### Decision · Program_Chairs · 2022-01-20

**Decision:**

Accept (Poster)

**Comment:**

This paper proposes a few-shot (untargeted) backdoor attack (FSBA) against siamese network-based visual object tracking. Contributions can be summarized as follows: First, this paper treats the attack task as an instance of multi-task learning and can be regarded as the first backdoor attack against VOT. Besides, a simple yet effective few-shot untargeted backdoor attack is proposed and achieves significant effectiveness in both digital and physical-world scenarios. This paper reveals the vulnerability of VOT to backdoor attacks caused by outsourced training or using third-party pre-trained models. One weakness is that threat model requires a very strong attacker with ability to modify the training algorithm, but only very simple defenses are considered. Overall, this is a good first attempt at showing vulnerability of VOT approaches.